# Efficient Preference Poisoning Attack on Offline RLHF

**Chenye Yang** [1]  **Weiyu Xu** [2]  **Lifeng Lai** [1]

## Abstract

Offline Reinforcement Learning from Human Feedback (RLHF) pipelines such as Direct Preference Optimization (DPO) train on a pre-collected preference dataset, which makes them vulnerable to preference poisoning attack. We study label flip attacks against log-linear DPO. We first illustrate that flipping one preference label induces a parameter-independent shift in the DPO gradient. Using this key property, we can then convert the targeted poisoning problem into a structured binary sparse approximation problem. To solve this problem, we develop two attack methods: Binary-Aware Lattice Attack (BAL-A) and Binary Matching Pursuit Attack (BMP-A). BAL-A embeds the binary flip selection problem into a binary-aware lattice and applies Lenstra-Lenstra-Lovász reduction and Babai's nearest plane algorithm; we provide sufficient conditions that enforce binary coefficients and recover the minimum-flip objective. BMP-A adapts binary matching pursuit to our non-normalized gradient dictionary and yields coherence-based recovery guarantees and robustness (impossibility) certificates for $K$-flip budgets. Experiments on synthetic dictionaries and the Stanford Human Preferences dataset validate the theory and highlight how dictionary geometry governs attack success.

## 1. Introduction

Reinforcement learning from human feedback (RLHF) is a popular approach for aligning a learned policy with human preferences when an explicit reward function is hard to specify (Christiano et al., 2017; Casper et al., 2023). In many

---

[1]Department of Electrical and Computer Engineering, University of California, Davis, Davis, CA, USA [2]Department of Electrical and Computer Engineering, University of Iowa, Iowa City, IA, USA. Correspondence to: Chenye Yang <cyyyang@ucdavis.edu>, Weiyu Xu <weiyu-xu@uiowa.edu>, Lifeng Lai <lflai@ucdavis.edu>.

*Proceedings of the $43^{rd}$ International Conference on Machine Learning*, Seoul, South Korea. PMLR 306, 2026. Copyright 2026 by the author(s).

practical pipelines, the policy is trained from a pre-collected dataset of pairwise preference labels (Christiano et al., 2017; Bai et al., 2022; Rafailov et al., 2023), which is referred to as *offline* RLHF (Xiong et al., 2024), to differentiate it from *online* RLHF where the policy is learned from real-time interactions with human feedbacks (Wang et al., 2023). Recent work has demonstrated that RLHF faces substantial security issues related to the reliability of the human preference data, i.e., when there are adversarial attacks on preferences or random noises in human preferences (Casper et al., 2023). The preference data can be manipulated by adversaries to mislead both offline and online RLHF to produce targeted outcomes (Baumgärtner et al., 2024; Wu et al., 2025a; Fu et al., 2025; Nika et al., 2025; Pathmanathan et al., 2025; Yang et al., 2025).

Two types of adversarial attack models are commonly studied in offline RLHF (Baumgärtner et al., 2024; Rando & Tramèr, 2024; Wu et al., 2025a; Nika et al., 2025): (i) label flip attack, which flips some labels within the original dataset; and (ii) data injection attack, which appends poisoned trajectories and preferences to the original dataset. In particular, (Rando & Tramèr, 2024; Baumgärtner et al., 2024; Wu et al., 2025a) provide valuable experimental results on the impact of label flip and data injection attacks, but focus less on the theoretical analysis. (Nika et al., 2025) provides a comprehensive theoretical analysis of the effectiveness of data injection attacks, showing that to mislead offline RLHF, the attacker needs to append numerous poisoned preference pairs, with size scaling linearly with the original dataset.

However, while label flip attacks on offline RLHF is practically plausible, a comprehensive theoretical study of such a setup remains an open problem. We aim to address this critical gap in this paper. Understanding this vulnerability is crucial for developing robust algorithms and ensuring the reliability of RLHF systems in real-world applications, e.g., designing helpful and harmless LLMs robust to adversarial attacks on training set. Compared with the data injection attack, the problem of understanding label flipping attack is more challenging: First, flips are limited to the original dataset: the attacker can only change labels on existing comparisons and cannot freely create arbitrary many new preference pairs. Second, the effect of flipping a single label on the learned policy is hard to predict, which makes it

difficult to effectively choose which labels to attack.

To our best knowledge, this work is the first to provide a theoretical analysis of *label flip* attacks on offline RLHF, specifically on Direct Preference Optimization (DPO) pipeline (Rafailov et al., 2023) under the log-linear policy class. Our focus on log-linear policies follows prior RLHF theory (Xiong et al., 2024; Chowdhury et al., 2024; Nika et al., 2025) and isolates the core geometric phenomenon that drives flip attacks. The main technical observation is simple but powerful: in log-linear DPO, flipping one preference label changes the gradient of the DPO objective by a vector that does not depend on the current policy parameter. The attacker's problem can therefore be expressed in *gradient space* as selecting a subset of existing preference pairs whose flip-induced gradient shifts add up to (or closely approximate) a target vector, which is decided by the attacker's desired policy to force the learner to adopt. We further show that controlling the gradient residual is a sufficient surrogate for policy-space attack success.

The reduction above implies that finding the optimal targeted flip attacks is a combinatorial problem and, in general, is computationally hard. To address this, we develop two algorithms with different focuses. First, for the general minimum-flip objective, we propose a *Binary-Aware Lattice Attack* (BAL-A). BAL-A uses a lattice embedding that simultaneously penalizes residual error and non-binary coefficients, and then applies Lenstra-Lenstra-Lovász (LLL) basis reduction and Babai's nearest-plane algorithm to produce an integer solution that can be interpreted as a flip pattern. We further analyze how the embedding parameter controls the method and show that sufficiently large penalty enforces small integer coefficients and the proposed method finds the true minimum-flip solution under a separation condition. Second, when the attacker has a flip budget $K$ and the true attack is sparse, we propose *Binary Matching Pursuit Attack* (BMP-A), a greedy method adapted from Binary Matching Pursuit (Wen & Li, 2021). BMP-A selects examples using normalized correlations to handle non-normalized gradient columns and admits coherence-based recovery guarantees. We also provide simple robustness certificates showing when *no $K$-flip attack* can succeed.

We validate the theory on synthetic dictionaries and on dictionaries constructed from the Stanford Human Preferences (SHP) dataset (Ethayarajh et al., 2022). The synthetic setting allows controlled tests of the lattice penalty and the coherence conditions. On SHP, we observe that BAL-A's success depends sharply on the penalty region predicted by the separation analysis, and that BMP-A benefits substantially from reducing dictionary coherence, supporting our theoretical results.

**Additional Related Work**: Adversarial attacks on bandit and standard RL problems have been extensively studied. In these setups, the attacker manipulates rewards, actions, and transitions to force the learner toward targeted behaviors. However, attacks on RLHF changes *human feedback*, such as preference labels, which are discrete, and indirectly shape the learned policy, making the effect of a single corruption harder to analyze. We summarize these additional related work in Appendix A.

## 2. Problem Formulation

### 2.1. Offline RLHF and DPO

In offline RLHF, we observe a fixed dataset $\mathcal{D}$ consisting of pairwise preferences over two candidate actions/responses $a$ under the same state/prompt $s$. We write each comparison as $(s, a, a', o)$, where $o \in \{+1, -1\}$ encodes which action is preferred by the human labeler. $o = +1$ means $a \succ a'$, while $o = -1$ means $a' \succ a$. Given a reference policy $\mu$ with parameter $\theta_\mu$, Direct Preference Optimization (DPO) trains a parametric policy $\pi_\theta$ directly from these comparisons, without explicitly fitting a reward model (Rafailov et al., 2023).

We focus on the log-linear policies for DPO, as defined in Definition 2.1. Log-linear policy class is commonly considered in RLHF literature (Xiong et al., 2024; Chowdhury et al., 2024; Nika et al., 2025), and it features many good properties in the further analysis.

**Definition 2.1** (Log-linear Policy (Nika et al., 2025)). Let $\psi$ be a $d$-dimensional feature mapping $\psi : \mathcal{S} \times \mathcal{A} \to \mathbb{R}^d$ with $\max_{s,a} \|\psi(s, a)\| \leq 1$. We consider the following class of log-linear policies $\forall (s, a) \in \mathcal{S} \times \mathcal{A}$ where $\theta \in \mathbb{R}^d$:

$$\Pi^{\log} = \left\{ \pi_\theta : \pi_\theta(a \mid s) = \frac{\exp\left(\psi(s, a)^\top \theta\right)}{\sum_{a'} \exp\left(\psi(s, a')^\top \theta\right)} \right\}.$$

Then, the DPO objective can be written as a regularized logistic loss over comparisons (Rafailov et al., 2023):

$$L_{\text{DPO}}(\theta; \mathcal{D}) = - \sum_{(s,a,a',o) \in \mathcal{D}} \log \sigma \left( o \cdot \beta \left( \log \frac{\pi_\theta(a \mid s)}{\mu(a \mid s)} \right. \right.$$
$$\left. \left. - \log \frac{\pi_\theta(a' \mid s)}{\mu(a' \mid s)} \right) \right) + \frac{\lambda}{2} \|\theta - \theta_\mu\|_2^2,$$

where $\sigma(\cdot)$ is the sigmoid function, $\beta > 0$ is a temperature parameter, and $\lambda > 0$ is the regularization parameter. For reader's convinience, we summarize the main steps of the derivation of DPO from the RLHF pipeline provided in (Rafailov et al., 2023) in Appendix B.

### 2.2. Preference Attack

We study label flip attacks on offline preference datasets. The attacker selects a subset of comparisons and flips their

outcomes, equivalently changing $o_i$ to $-o_i$ on the chosen indices. The size of the dataset is fixed. Let $x \in \{0,1\}^n$ indicate which comparisons are flipped, and write $\tilde{\mathcal{D}}(x)$ as the attacked dataset.

In this paper, a targeted label flip attacker's optimization problem is formulated as:

$$\min_{x \in \{0,1\}^n} \|x\|_0 \quad \text{s.t.} \quad \left\|\pi_{\hat{\theta}(\tilde{\mathcal{D}}(x))} - \pi^\dagger\right\|_1 \leq \varepsilon,$$

Here $\hat{\theta}(\tilde{\mathcal{D}}(x))$ is the DPO policy parameter trained on the attacked dataset, and $\pi_{\hat{\theta}(\tilde{\mathcal{D}}(x))}$ is the corresponding policy. In this formulation, the goal of the attacker is to force $\pi_{\hat{\theta}(\tilde{\mathcal{D}}(x))}$ to align with a target policy $\pi^\dagger$ (with parameter $\theta^\dagger$) decided by the attacker, while modifying as few labels as possible.

We also consider a budgeted variant, in which the attacker can at most flip $K$ entries in the training dataset:

$$\min_{x \in \{0,1\}^n:\, \|x\|_0 \leq K} \|x\|_0 \quad \text{s.t.} \quad \left\|\pi_{\hat{\theta}(\tilde{\mathcal{D}}(x))} - \pi^\dagger\right\|_1 \leq \varepsilon,$$

## 3. Label Flip Attack on DPO

### 3.1. Effect of Label Flip Attack

The DPO optimization problem with log-linear policy can be written as $\min_\theta L_{\mathrm{DPO}}(\theta; \mathcal{D})$. For sample $i$: $(s_i, a_i, a_i', o_i)$, the per-sample loss is

$$\ell_i(\theta) = -\log \sigma\left(o_i\, \beta\left[\log \frac{\pi_\theta(a_i|s_i)}{\mu(a_i|s_i)} - \log \frac{\pi_\theta(a_i'|s_i)}{\mu(a_i'|s_i)}\right]\right). \tag{1}$$

Using (1), the overall DPO loss can be written as:

$$L_{\mathrm{DPO}}(\theta; \mathcal{D}) = \sum_{j \in \mathcal{D}} \ell_j(\theta) + \frac{\lambda}{2}\|\theta - \theta_\mu\|^2.$$

**Theorem 3.1** (Flip-induced gradient shift for log-linear DPO)**.** *Fix a comparison $(s_i, a_i, a_i', o_i)$ with $o_i \in \{+1, -1\}$, and define the feature difference $\Delta\psi_i := \psi(s_i, a_i) - \psi(s_i, a_i') \in \mathbb{R}^d$. Let $\ell_i(\theta)$ be the per-sample DPO loss in (1), and let $g_i(\theta) := \nabla_\theta \ell_i(\theta)$. If we flip the label $o_i$ to $\tilde{o}_i = -o_i$, and we have $\tilde{g}_i(\theta)$ replacing the $o_i$ with $\tilde{o}_i$ in $g_i(\theta)$, then for every $\theta$ the per-sample gradient changes by a constant vector that is independent of $\theta$:*

$$\Delta g_i := \tilde{g}_i(\theta) - g_i(\theta) = o_i\, \beta\, \Delta\psi_i.$$

Theorem 3.1 is the key structural property behind our attack formulation: each flipped comparison contributes a fixed "atom" in gradient space for all values of policy parameter $\theta$. Appendix C formalizes two related extension identities: DPO with general differentiable policy classes beyond the log-linear class has an exact but generally parameter-dependent flip-induced gradient shift, and the

reward-modeling phase of RLHF has a similar parameter-independent identity for linear rewards. These identities show the potential generality of our attack formulation to other DPO policy classes and RLHF pipelines.

### 3.2. Formulate Label Flip Attack Problem

Using Theorem 3.1, when the attacker flip a set $\mathcal{F} \subset \mathcal{D}$, the attacked gradient becomes:

$$\nabla_\theta L_{\mathrm{DPO}}(\theta; \tilde{\mathcal{D}}) = \sum_{j \notin \mathcal{F}} g_j(\theta) + \sum_{i \in \mathcal{F}} \tilde{g}_i(\theta) + \lambda(\theta - \theta_\mu)$$
$$= \nabla_\theta L_{\mathrm{DPO}}(\theta; \mathcal{D}) + \Delta g_\mathcal{F},$$

where $\Delta g_\mathcal{F} := \sum_{i \in \mathcal{F}} \Delta g_i$ is a term depending on $\mathcal{F}$, but not on $\theta$. Strong convexity of $L_{\mathrm{DPO}}(\theta; \tilde{\mathcal{D}})$ (Lemma E.14 (Nika et al., 2025)) ensures that the first order condition (FOC) is enough to guarantee optimality. Then FOC minimizer for the attacked problem, $\tilde{\theta}$, satisfies:

$$\nabla_\theta L_{\mathrm{DPO}}(\tilde{\theta}; \tilde{\mathcal{D}}) = \nabla_\theta L_{\mathrm{DPO}}(\tilde{\theta}; \mathcal{D}) + \Delta g_\mathcal{F} = 0.$$

Now we need to study: if we want the learned policy $\tilde{\theta}$ to be some (target) policy $\theta^\dagger$, what should the set $\mathcal{F}$ be?

From the FOC above, we have:

$$g^\dagger := \nabla_\theta L_{\mathrm{DPO}}(\theta^\dagger; \mathcal{D}) = -\Delta g_\mathcal{F} = -\sum_{i \in \mathcal{F}} o_i \beta\, \Delta\psi_i,$$

where we define $g^\dagger$ as the gradient of the clean DPO loss at the target $\theta^\dagger$.

For simplicity, denote $v_i = o_i\beta\Delta\psi_i$, $V = [v_1, \ldots, v_n] \in \mathbb{R}^{d \times n}$. Then the label flip attack problem becomes:

$$\min_\mathcal{F} \quad |\mathcal{F}| \quad \text{s.t.} \quad \sum_{i \in \mathcal{F}} v_i = -g^\dagger.$$

It can be seen as a binary sparse approximation problem:

$$\min_{x \in \{0,1\}^n} \quad \mathbf{1}^\top x \quad \text{s.t.} \quad Vx = -g^\dagger.$$

We note that, in the label flip attack problem, the matrix $V$ is fixed and determined by the original dataset $\mathcal{D}$, and the target gradient $-g^\dagger$ is also fixed and determined by the target policy $\theta^\dagger$. The vector $x$ is a binary vector indicating which samples to flip in the dataset $\mathcal{D}$.

When we allow some approximation error $\varepsilon$ in reaching the target policy $\theta^\dagger$, we say that: a flip attack is *exactly successful* if $Vx + g^\dagger = 0$, and *approximately successful* if $\|Vx + g^\dagger\|_2 \leq \varepsilon$. Thus, the exact and approximate flip attack problems can be written as:

$$\min_{x \in \{0,1\}^n} \quad \mathbf{1}^\top x \quad \text{s.t.} \quad Vx + g^\dagger = 0, \tag{2}$$

and

$$\min_{x \in \{0,1\}^n} \quad \mathbf{1}^\top x \quad \text{s.t.} \quad \|Vx + g^\dagger\|_2 \leq \varepsilon. \tag{3}$$

### 3.3. From Gradient Residual to Policy Closeness

For log-linear policies, prior work justifies treating a *parameter-space* target as a surrogate for a *policy-space* success criterion. In particular, Lemma E.5 of (Nika et al., 2025) relate an $\ell_1$ policy-closeness constraint with tolerance $\epsilon$ to a parameter-closeness constraint with tolerance $\epsilon'$: when $\epsilon' \leq \epsilon/(2\sqrt{d})$, parameter feasibility implies policy feasibility, and they also provide conditions for the reverse implication. Motivated by this equivalence, we analyze poisoning around a target parameter $\theta^\dagger$ and connect attack success to how flips perturb the DPO optimality conditions.

By construction,

$$\nabla_\theta L_{\text{DPO}}(\theta^\dagger; \tilde{\mathcal{D}}) = \nabla_\theta L_{\text{DPO}}(\theta^\dagger; \mathcal{D}) + Vx = g^\dagger + Vx. \tag{4}$$

Hence, the approximate flip-attack constraint

$$\|Vx + g^\dagger\|_2 \leq \varepsilon \tag{5}$$

is exactly a bound on the *gradient residual* at $\theta^\dagger$ for the poisoned objective, i.e., it enforces that $\theta^\dagger$ is an $\varepsilon$-stationary point of $L_{\text{DPO}}(\,\cdot\,; \tilde{\mathcal{D}})$. The following lemma shows that, under strong convexity, such approximate stationarity guarantees that training on $\tilde{\mathcal{D}}$ produces a parameter (and thus a policy) close to the target.

**Lemma 3.2** (Gradient residual to policy closeness). *Assume $L_{\text{DPO}}(\,\cdot\,; \tilde{\mathcal{D}})$ is differentiable and $m$-strongly convex in $\theta$, e.g., for log-linear DPO with $\ell_2$ regularization, see (Nika et al., 2025). Let $\hat{\theta} := \arg\min_\theta L_{\text{DPO}}(\theta; \tilde{\mathcal{D}})$ and define the trained policy $\pi_{\theta,\tilde{\mathcal{D}}} := \pi_{\hat{\theta}}$, with target policy $\pi^\dagger := \pi_{\theta^\dagger}$. If $\|Vx + g^\dagger\|_2 \leq \varepsilon$ holds, equivalently $\|\nabla_\theta L_{\text{DPO}}(\theta^\dagger; \tilde{\mathcal{D}})\|_2 \leq \varepsilon$, then*

$$\|\hat{\theta} - \theta^\dagger\|_2 \leq \varepsilon/m. \tag{6}$$

*Moreover, for log-linear policies this parameter bound implies the policy-space guarantee $\|\pi_{\hat{\theta}} - \pi^\dagger\|_1 \leq \epsilon$ whenever $\varepsilon \leq m\,\epsilon/(2\sqrt{d})$.*

Lemma 3.2 justifies (5) as a sufficient condition for approximate attack success in policy space. This motivates our focus on the gradient perturbation at $\theta^\dagger$; in the log-linear DPO case, the per-example flip induces a parameter-independent gradient shift, yielding a fixed dictionary of vectors (the columns of $V$) and the binary selection formulation developed before.

### 3.4. Feasibility of Label Flip Attack

The feasibility question asks whether there exists $x \in \{0,1\}^n$ such that $\|Vx + g^\dagger\|_2 \leq R$, where $R = 0$ for (2) and $R = \varepsilon$ for (3). This decision problem is NP-hard in general. Equivalently, feasibility means that $-g^\dagger$ lies within distance $R$ of the finite attainable set $\mathcal{A} := \{Vx : x \in \{0,1\}^n\}$.

A target $\pi^\dagger$ closer to the clean optimum $\pi^\star$ generally makes the attack easier because $|g^\dagger|$ is typically smaller (exactly

zero if $\pi^\dagger = \pi^\star$), but closeness alone is not sufficient: feasibility depends not only on $\pi^\dagger$, but also on the geometry of the dictionary $V$, i.e., whether $-g^\dagger$ can be well approximated by a sparse binary combination of $V$'s columns.

Therefore, we assume in the following that a feasible flip attack exists as in Assumption 3.3, and focus on finding the attack set $\mathcal{F}$ and quantifying the cost of such an attack in terms of the number of flipped labels $|\mathcal{F}|$.

**Assumption 3.3** (Existence of feasible flip attack). There exists a *true* binary attack vector $x^\star \in \{0,1\}^n$ such that $\|Vx^\star + g^\dagger\|_2 \leq R$. In the exact problem we take $R = 0$, and in the approximate problem we take $R = \varepsilon$.

### 3.5. Lower-bound of Flips

We now derive a simple lower bound on the number of flipped labels $|\mathcal{F}|$ required for a successful flipping attack.

**Theorem 3.4** (Norm-based lower-bound of $|\mathcal{F}|$). *If the attacker achieves the exact constraint in the exact problem (2), then the number of flips ($|\mathcal{F}|$ or $\mathbf{1}^\top x$) must satisfy $|\mathcal{F}| \geq \|g^\dagger\|_2/(2\beta)$. If the attacker is allowed tolerance $\varepsilon \geq 0$ and achieves the constraint in the approximate problem (3), then the number of flips must satisfy $|\mathcal{F}| \geq (\|g^\dagger\|_2 - \varepsilon)/(2\beta)$.*

Theorem 3.4 shows that, the number of flips must grow at least linearly with the norm of the target gradient $\|g^\dagger\|_2$. The bounds do not require any structural assumptions on the attack matrix $V$ beyond the feature norm bound from Definition 2.1. They provide a simple necessary condition on $|\mathcal{F}|$ under the Assumption 3.3 of a feasible flip attack. The dependence on $\beta$ comes from the per-flip shift $o_i\beta\Delta\psi_i$, of which the norm is proportional to $\beta$. Thus, smaller $\beta$ makes each individual label flip weaker in gradient space, so more flips may be needed to cancel a fixed $g^\dagger$.

## 4. Binary-Aware Lattice Attack (BAL-A)

In this section, we discuss why a standard lattice formulation for the closest vector problem (CVP) can not handle the flip attack problem, even they are similar. Motivated by the limitations, we introduce a *binary-aware* lattice embedding combined with Babai's nearest-plane algorithm. We derive sufficient conditions under which the result is guaranteed to be binary and recover the minimum-flip.

### 4.1. Lattice Method and a Binary-aware Embedding

The exact flip attack is the minimum-flip feasibility problem:

$$\min_{x \in \{0,1\}^n} \mathbf{1}^\top x \quad \text{s.t.} \quad Vx + g^\dagger = 0. \tag{7}$$

A common approximate relaxation approach replaces the hard constraint by residual minimization, which, however,

results in a different objective:

$$\min_{x \in \{0,1\}^n} \|Vx + g^\dagger\|_2. \tag{8}$$

A natural relaxation of (8) is to allow integer coefficients,

$$\min_{z \in \mathbb{Z}^n} \|Vz + g^\dagger\|_2, \tag{9}$$

and view this as a CVP in the lattice (Lenstra et al., 1982) $L := \{Vz : z \in \mathbb{Z}^n\} \subset \mathbb{R}^d$ with target $t := -g^\dagger$. Indeed, $\text{dist}(L, t) := \min_{z \in \mathbb{Z}^n} \|Vz - t\|_2 = \min_{z \in \mathbb{Z}^n} \|Vz + g^\dagger\|_2$. One can then apply lattice basis reduction (e.g., LLL (Lenstra et al., 1982)) and Babai's nearest-plane algorithm (Babai, 1986) to obtain an integer vector $z \in \mathbb{Z}^n$ that approximately solves (9).

The lattice relaxation (9) is a natural starting point, but it does not directly match the binary minimum-flip model (7) in two key ways: binary coefficients, and minimum-flip objective. We summarize the mismatches in Appendix D.

These issues motivate a different design: rather than solving an unrestricted integer problem and truncating afterwards, we construct a *binary-aware* lattice embedding where non-binary coefficients are heavily penalized in the lattice norm, so that any sufficiently short lattice vector must already correspond to a (near) binary solution.

We work directly with the real-valued attack matrix $V$ and gradient $g^\dagger$. Assume that the flip-effect vectors are uniformly bounded $\|v_i\|_2 \le B$ ($B = 2\beta$, shown in Appendix J.3), we build a lattice in $\mathbb{R}^{d+n}$ using the $(d+n) \times (n+1)$ real basis

$$B_{\text{bin}} := \begin{pmatrix} V & -g^\dagger \\ MI_n & 0 \end{pmatrix}, \qquad M > 0,$$

whose columns are $b_i = (v_i; Me_i)$ for $i = 1, \dots, n$ and $b_{n+1} = (-g^\dagger; 0)$. For any integer vector $z \in \mathbb{Z}^n$, consider the coefficient vector $u(z) := (z; -1) \in \mathbb{Z}^{n+1}$, and the corresponding lattice vector

$$y(z) := B_{\text{bin}} u(z) = \begin{pmatrix} Vz + g^\dagger \\ Mz \end{pmatrix} \in \mathbb{R}^{d+n}.$$

Its squared norm decomposes as

$$\|y(z)\|_2^2 = \|Vz + g^\dagger\|_2^2 + M^2 \|z\|_2^2. \tag{10}$$

Intuitively, the top block of $B_{\text{bin}}$ measures how well $z$ approximates the target $-g^\dagger$ (attack effectiveness), while the bottom block penalizes the magnitude of the coefficients $z_i$. Thus, choosing $M$ sufficiently large discourages non-physical solutions with $|z_i| \ge 2$, addressing the first mismatch above. Moreover, for binary flip indicators $x \in \{0,1\}^n$ we have $\|x\|_2^2 = \mathbf{1}^\top x$, so restricting (10) to binary vectors yields

$$\min_{x \in \{0,1\}^n} \|Vx + g^\dagger\|_2^2 + M^2 \mathbf{1}^\top x, \tag{11}$$

---

**Algorithm 1** Binary-Aware Lattice Attack (BAL-A)

**input** $V = [v_1, \dots, v_n] \in \mathbb{R}^{d \times n}$, $g^\dagger \in \mathbb{R}^d$, $M > 0$, $\delta \in (1/4, 1)$
**output** $\hat{x} \in \{0,1\}^n$
1: Set $b_i := (v_i; Me_i)$, $B_e := [b_1, \dots, b_n]$, and $t := (-g^\dagger; 0)$.
2: Compute an LLL-reduced basis $(\tilde{B}, T) \leftarrow \text{LLL}_\delta(B_e)$ with $\tilde{B} = B_e T$.
3: Run Babai nearest-plane decoding on $(\tilde{B}, t)$ and obtain coefficients $z$.
4: Map back to the original embedding coordinates: $x_{\text{int}} \leftarrow Tz$.
5: $\hat{x}_i := \min\{1, \max\{0, x_{\text{int},i}\}\}$ for each $i$.

---

which couples residual minimization with the flip count and hence address the second mismatch. In particular, on the feasible set $\{x \in \{0,1\}^n : Vx + g^\dagger = 0\}$ the residual term vanishes and (11) reduces to minimizing $\mathbf{1}^\top x$; therefore, whenever an exact attack is achievable, minimizing $\|y(x)\|_2$ over feasible binary solutions is equivalent to (2).

We summarize the practical Binary-Aware Lattice Attack (BAL-A), which combines lattice embedding with Babai's nearest-plane algorithm, in Algorithm 1. The inputs are the dataset matrix $V \in \mathbb{R}^{d \times n}$, the target gradient $g^\dagger \in \mathbb{R}^d$, and a penalty parameter $M > 0$. The method first embeds the binary flip selection problem into a lattice by augmenting each column $v_i$ with a scaled identity component $Me_i$, so that nearest-lattice decoding corresponds to minimizing the binary-aware objective $|Vz + g^\dagger|_2^2 + M^2 |z|_2^2$ over integer vectors $z$. It then applies LLL reduction to obtain a better-conditioned basis and uses Babai's nearest-plane rounding to recover an integer coefficient vector, which is mapped back to the original coordinates and then truncated to produce a candidate flip pattern. For completeness, we provide the full step-by-step procedure (including the LLL and Babai) in Appendix E.

From a computational perspective, the dominant cost of BAL-A is the lattice preprocessing on the $(d+n) \times n$ embedded basis, that is, LLL reduction plus the linear-algebra preprocessing used by Babai. Babai itself is inexpensive once the reduced basis is available, but the preprocessing cost grows quickly with $n$, which is why in our experiments BAL-A is used only for small-to-moderate attacked subsets.

### 4.2. Theoretical Guarantees for BAL-A

#### 4.2.1. LARGE $M$ ENFORCES BINARY COEFFICIENTS

We formalize how the penalty parameter $M$ controls the integer coefficients in the binary-aware embedding.

**Lemma 4.1** (Coefficient bound for the shortest binary-aware lattice vector). *Assume the flip-effect vectors are bounded as*

$\|v_i\|_2 \leq B$ for all $i$. Suppose there exists a binary flip attack $x^\star \in \{0,1\}^n$ with support size $K^\star := \|x^\star\|_0$ and residual $\|Vx^\star + g^\dagger\|_2 \leq R$. Let $z^{\mathrm{opt}} \in \arg\min_{z \in \mathbb{Z}^n} \|y(z)\|_2$ be any minimizer of the binary-aware lattice norm over $\mathbb{Z}^n$. Then there exists $M_0 = M_0(B, R, K^\star)$ such that for all $M \geq M_0$, every coordinate satisfies $|z_i^{\mathrm{opt}}| \leq 1$, i.e., $z^{\mathrm{opt}} \in \{-1, 0, 1\}^n$. One explicit sufficient choice is

$$M_0 = (B\sqrt{K^\star} + \sqrt{B^2 K^\star + 6BR + 3B^2})/3. \quad (12)$$

Lemma 4.1 shows that if there exists a reasonably good binary attack $x^\star \in \{0,1\}^n$, then for a sufficiently large $M$ the global minimizer of the embedded lattice norm over $\mathbb{Z}^n$ cannot contain any coefficient with magnitude $\geq 2$. We do not claim the explicit threshold in (12) is tight; its role is to exhibit a provable binary-enforcing method, and the experiments confirm that it can be conservative in practice.

**Theorem 4.2** (Binary optimality under nonnegativity). *Under the assumptions of Lemma 4.1, assume additionally that we restrict to nonnegative coefficients, i.e., $z \in \mathbb{Z}_{\geq 0}^n$, which is natural when each $v_i$ already encodes the flip direction. Let $z^{\mathrm{opt}} \in \arg\min_{z \in \mathbb{Z}_{\geq 0}^n} \|y(z)\|_2$. If $M \geq M_0(B, R, K^\star)$, then $z^{\mathrm{opt}} \in \{0,1\}^n$ and*

$$\|y(z^{\mathrm{opt}})\|_2^2 = \|Vz^{\mathrm{opt}} + g^\dagger\|_2^2 + M^2\|z^{\mathrm{opt}}\|_2^2$$
$$\leq \|y(x^\star)\|_2^2 \leq R^2 + M^2 K^\star.$$

Theorem 4.2 converts the coefficient bound from Lemma 4.1 into an actual $\{0,1\}^n$ guarantee under the natural constraint $z \in \mathbb{Z}_{\geq 0}^n$. Thus, the optimizer of the penalized embedding is a valid flip attack, which addresses the first mismatch.

#### 4.2.2. SMALL $M$ RECOVERS MINIMUM-FLIP OBJECTIVE

We now analyze how $M$ controls the recovery of the minimum-flip objective in our flip attack problem.

**Theorem 4.3** (Minimum-residual recovers minimum-flip). *Assume the exact flip attack is feasible, and let $x^\star \in \arg\min_{x \in \{0,1\}^n} \mathbf{1}^\top x$ s.t. $Vx + g^\dagger = 0$. Define $K^\star := \mathbf{1}^\top x^\star$. For each $k \in \{0, 1, \ldots, K^\star - 1\}$ define the best $k$-flip residual $\rho_k := \min_{x \in \{0,1\}^n: \mathbf{1}^\top x = k} \|Vx + g^\dagger\|_2$. Consider the binary-aware surrogate objective*

$$F_M(x) := \|Vx + g^\dagger\|_2^2 + M^2 \mathbf{1}^\top x, \quad x \in \{0,1\}^n.$$

*If $M$ satisfies the separation condition*

$$\rho_k^2 > M^2 (K^\star - k) \quad \text{for all } k = 0, 1, \ldots, K^\star - 1, \quad (13)$$

*then every minimizer of $\min_{x \in \{0,1\}^n} F_M(x)$ is an optimal exact attack pattern; in particular, any minimizer is feasible $(Vx + g^\dagger = 0)$ and has $\mathbf{1}^\top x = K^\star$.*

Theorem 4.3 states that $F_M(x)$ recovers the minimum-flip in exact attack case if all under-budget patterns ($k < K^\star$)

satisfy separation condition, and any minimizer of $F_M$ must be feasible and use exactly $K^\star$ flips. This can be seen by a counterexample that when $M \geq \|g^\dagger\|_2$, the zero-flip solution $x = 0$ achieves the smallest objective value than any feasible solution with at least one flip. We note that the exact separation condition in Theorem 4.3 involves combinatorial computation and is intended only as a sufficient condition for theoretical validation. It is not used as part of the scalable BAL-A pipeline.

The surrogate objective $F_M$ trades off two goals: reducing the residual $\|Vx + g^\dagger\|_2$ and promoting discrete binary coefficients. In general, these goals can conflict, so there is no single, universal choice of $M$ that guarantees recovering a feasible minimum-flip attack pattern for all instances. Moreover, if $M$ is chosen too large, the penalty can favor under-selection (too few flips) even when an exact solution exists. Formal counterexamples are given in Appendix F.

## 5. Binary Matching Pursuit Attack (BMP-A)

In this section, we introduce the attack budget constraint $K$ and study the sparse $K$-flip attack model. Then the $K$-flip attack problem can be viewed as a sparse recovery problem with a binary constraint on the coefficients. We adapt the Binary Matching Pursuit (BMP) algorithm (Wen & Li, 2021) to our flip attack setting and further provide sufficient conditions under which no $K$-flip attack can succeed.

### 5.1. Sparse $K$-Flip Attack Model

Equivalent to the flip attack formulation in Section 3.4, we define the minimum number of flips required for approximate success as

$$K^\star := \min_{x \in \{0,1\}^n} \mathbf{1}^\top x \quad \text{s.t.} \quad \|Vx + g^\dagger\|_2 \leq R, \quad (14)$$

and ask whether $K^\star < \infty$ (i.e., whether the constraint set is nonempty). Problem (14) is a binary optimization problem and is NP-hard in general.

As in classical sparse recovery, our analysis assumes that there exists an underlying feasible sparse solution to the attack constraint.

**Assumption 5.1** (Existence of a feasible $K^\star$-flip attack). There exists a *true* binary attack vector $x^\star \in \{0,1\}^n$ and an integer $K^\star \geq 1$ such that $\|x^\star\|_0 = K^\star$ and $\|Vx^\star + g^\dagger\|_2 \leq R$. In the exact problem we take $R = 0$, and in the approximate problem we take $R = \varepsilon$.

In practice, an adversary is typically constrained to modify only a small fraction of the data. We therefore focus on the *sparse attack* regime, where $K^\star$ is small. This motivates a budgeted $K$-flip model: for a given budget $K \geq K^\star$, we restrict attention to attacks with $\|x\|_0 \leq K$ and study (i) impossibility conditions under which no such sparse attack

can succeed, and (ii) algorithmic recovery of a successful sparse attack when it exists. Formally, we assume

**Assumption 5.2** (Sparse attack model). $\|x^\star\|_0 \leq K$ (equivalently, $K^\star \leq K$).

In this way, $K$ simultaneously models the attacker's budget and provides a robustness scale: larger $K$ makes attacks easier, while small $K$ may render attacks impossible.

### 5.2. BMP-A's Connection to Sparse Recovery

Our sparse flip-attack formulation is closely related to classical sparse recovery / sparse approximation. In the standard sparse recovery model, one observes (Tropp & Gilbert, 2007; Wen & Li, 2021) $y = Ax^\star + e$, where $A \in \mathbb{R}^{d \times n}$ is a dictionary (or sensing matrix), $x^\star \in \mathbb{R}^n$ is $K$-sparse, and $e$ models noise. The goal is to identify the sparse support (and possibly the coefficients) of $x^\star$ from $y$. A canonical greedy method is Orthogonal Matching Pursuit (OMP) (Tropp & Gilbert, 2007), which iteratively selects the column of $A$ most correlated with the current residual and then updates the residual after incorporating the selected atom.

In our setting, the "measurement" vector is the attack target $y := -g^\dagger$, the dictionary is the per-sample gradient contribution matrix $A := V$, and the attack is to find a *sparse binary* selector $x \in \{0,1\}^n$ such that $Vx \approx -g^\dagger$, i.e., $y$ is approximated by a sum of a small number of columns of $V$. Compared with classical sparse recovery, our coefficients are constrained to be binary (fixed to $+1$ on the chosen support), so the main task is *support selection* rather than continuous coefficient estimation. This viewpoint motivates pursuit-style greedy solvers, leading to our Binary Matching Pursuit Attack (BMP-A) in Algorithm 2.

In BMP-A, we adapt BMP (Wen & Li, 2021) to our setting while keeping the algorithm expressed in terms of the original non-normalized columns $v_i$ of $V$ and a binary selection vector $x \in \{0,1\}^n$. Starting from $r^0 = y$ and $x^0 = 0$, BMP-A greedily picks one *unselected* index using the normalized score $|\langle v_i, r \rangle| / \|v_i\|_2$ per iteration and updates the residual by subtracting the selected *original* column $r \leftarrow r - v_{i_t}$. It stops after $K$ iterations when the budget is exhausted, or when the residual is small $\|r\|_2 \leq \varepsilon$.

Most sparse-recovery analyses, including those for BMP (Wen & Li, 2021), assume that the dictionary columns are normalized to unit $\ell_2$-norm. In our problem, the columns $v_i$ of $V$ need not satisfy $\|v_i\|_2 = 1$, so BMP-A selects indices using *normalized* correlations. Define

$$a_i := \|v_i\|_2 > 0, \qquad u_i := \frac{v_i}{a_i},$$

$$D := \mathrm{diag}(a_1, \ldots, a_n), \quad U := [u_1, \ldots, u_n] = VD^{-1}.$$

Then for any binary attack $x \in \{0,1\}^n$, we have $Vx = Uz$ with $z := Dx$, so $z$ is $K$-sparse and has the *same sup-*

---

**Algorithm 2** Binary Matching Pursuit Attack (BMP-A)

**input** $V = [v_1, \ldots, v_n] \in \mathbb{R}^{d \times n}$, target $y = -g^\dagger$, budget $K$, tolerance $\varepsilon$
**output** $\hat{x} \in \{0,1\}^n$
1: $r \leftarrow y, \hat{x} \leftarrow 0, \Gamma \leftarrow \emptyset$.
2: **for** $t = 1, 2, \ldots, K$ **do**
3:     $i_t \in \arg\max_{i \in [n] \setminus \Gamma} |\langle v_i, r \rangle| / \|v_i\|_2$.
4:     $\Gamma \leftarrow \Gamma \cup \{i_t\}, \hat{x}_{i_t} \leftarrow 1$, and $r \leftarrow r - v_{i_t}$.
5:     **if** $\|r\|_2 \leq \varepsilon$ **then**
6:         break
7:     **end if**
8: **end for**

---

*port* as $x$, but with nonzero amplitudes $\{a_i : x_i = 1\}$ rather than 1. This normalization explains the BMP-A selection rule: at each step, $i_t \in \arg\max_{i \in [n] \setminus \Gamma} \frac{|\langle v_i, r \rangle|}{\|v_i\|_2} = \arg\max_{i \in [n] \setminus \Gamma} |\langle u_i, r \rangle|$, which is the standard correlation criterion under the unit-norm dictionary $U$. Likewise, mutual coherence is naturally defined using normalized directions,

$$\mu(V) := \max_{i \neq j} \frac{|\langle v_i, v_j \rangle|}{\|v_i\|_2 \|v_j\|_2} = \max_{i \neq j} |\langle u_i, u_j \rangle|.$$

Note, however, that passing from $(V, x)$ to $(U, z)$ changes the nonzero magnitudes from 1 to $a_i$; consequently, recovery guarantees stated for unit-norm dictionaries with equal-magnitude nonzeros require a mild adjustment that depends on the column-norm ratio.

### 5.3. Theoretical Guarantees for BMP-A

5.3.1. RECOVERY OF $x^\star$ WITHIN BUDGET $K$

Under standard binary sparse-recovery conditions, we specially adapt the following guarantees to our setting.

**Theorem 5.3** (Recovery under coherence). *Under the $K^\star$-flip Assumption 5.1. Let $\mu(V)$ be defined as above and $b := \min_{1 \leq i \leq n} \|v_i\|_2$, $B := \max_{1 \leq i \leq n} \|v_i\|_2$. If*

$$\mu(V) < \frac{b}{(2K^\star - 1)B} \tag{15}$$

*and*

$$\varepsilon < (b - (2K^\star - 1)\mu(V)B)/2, \tag{16}$$

*then BMP-A (Algorithm 2), selects an index in $\mathrm{supp}(x^\star)$ at each iteration. Consequently, after $K^\star$ iterations it exactly recovers $\mathrm{supp}(x^\star)$ and returns a binary vector $\hat{x}$ with $\mathrm{supp}(\hat{x}) = \mathrm{supp}(x^\star)$. Moreover, once the correct support has been selected, $r^{K^\star} = y - V\hat{x} = y - Vx^\star = e$, so the stopping rule $\|r\|_2 \leq \varepsilon$ triggers no later than iteration $K^\star$.*

Theorem 5.3 is the direct analog of the coherence condition in (Wen & Li, 2021), but adapted to non-normalized

columns through the norm range $b \leq \|v_i\|_2 \leq B$. Let $\rho := B/b$, the mutual coherence requirement scales as $\mu(V) \lesssim \frac{1}{(2K^\star - 1)\rho}$: highly non-uniform column norms (large $\rho$) make recovery more stringent. When $b = B$, e.g., after normalization, (15)–(16) reduce to the standard guarantee of (Wen & Li, 2021).

### 5.3.2. IMPOSSIBILITY CONDITIONS FOR $K$-FLIP ATTACK

We now give sufficient conditions under which *no $K$-flip attack can succeed*, independent of specific attack algorithms.

**Theorem 5.4** (Spectral norm and coherence impossibility). *Let $B := \max_i \|v_i\|_2$, $\mu(V)$ be defined as above, and $\|V\|_2$ be the spectral norm of $V$. Fix a budget $K \geq 1$ and tolerance $\varepsilon \geq 0$. If*

$$\|g^\dagger\|_2 - \varepsilon > \sqrt{K}\|V\|_2, \tag{17}$$

*or*

$$\left(\|g^\dagger\|_2 - \varepsilon\right)^2 > B^2\left(K + \mu(V)\, K(K-1)\right), \tag{18}$$

*then there does not exist any $x \in \{0,1\}^n$ with $\|x\|_0 \leq K$ such that $\|Vx + g^\dagger\|_2 \leq \varepsilon$. In particular, (17) is a spectral-norm certificate, while (18) is a coherence-based refinement.*

Theorem 5.4 provides sufficient conditions under which a $K$-flip attacker cannot drive the gradient residual at $\theta^\dagger$ below $\varepsilon$. It shows that the success of any label flip attack is limited by the geometry and scale of the flip-effect dictionary $V$. The spectral-norm condition (17) is universal: it only depends on the global operator norm $\|V\|_2$ and scales as $\sqrt{K}$. The coherence condition (18) is more refined: it separates the per-example strength $B$ from the directional similarity $\mu(V)$. When columns are weak (small $B$) and point in diverse directions (small $\mu(V)$), even the best choice of $K$ flips cannot create a large enough shift to cancel $g^\dagger$. For example, when $K = 1$, the coherence condition reduces to the simple no-single-flip condition $\|g^\dagger\|_2 - \varepsilon > B$. In the normalized case $B = 1$, it becomes $(\|g^\dagger\|_2 - \varepsilon)^2 > K + \mu(V)K(K-1)$, making explicit that larger budgets and more coherent dictionaries make attacks easier. These are certificates of robustness of DPO: failing them does not imply an attack exists, but satisfying them rules out *all $K$-flip attacks*, independent of specific attack algorithms.

## 6. Experiments and Results

In this section, we evaluate the proposed methods on both synthetic and real preference datasets.

### 6.1. Setup

We construct experiments with a ground-truth $K^\star$-flip attack pattern following Assumption 3.3. In each trial, we sample a support $S \subseteq \{1, \ldots, n\}$ with $|S| = K^\star$, set $x^\star = \mathbf{1}_S$,

and define the target vector as $t := Vx^\star = -g^\dagger$. Thus the exact attack constraint is feasible by construction. Given an output $\hat{x}$ by the attack algorithm BAL-A/BMP-A, we report the fraction of indices that are recovered against $x^\star$, i.e., true positive rate (TPR), and the residual $\|V\hat{x} - t\|_2$.

For synthetic data, we construct synthetic $V$ from normalized random Gaussian columns. To validate the BAL-A separation behavior, we use a dictionary with $V \in \mathbb{R}^{64 \times 20}$ and $K^\star = 5$ and sweep the lattice penalty $M$ over 25 log-spaced values; we apply LLL pre-reduction ($\delta = 0.75$) before Babai and run 200 Monte Carlo trials. To validate BMP-A in the sparse regime, we fix a single low-coherence dictionary $V \in \mathbb{R}^{200 \times 200}$ and sweep the true sparsity level $K^\star$; for each $K^\star$ we run 200 trials and run BMP-A for exactly $K^\star$ iterations.

For real data, we construct $V$ from the Stanford Human Preferences (SHP) dataset (Ethayarajh et al., 2022) by training a log-linear DPO model and forming the per-example flip-induced gradient shifts. BAL-A is evaluated on a smaller subset ($n = 50$, $K^\star = 7$) because of lattice reduction cost, while BMP-A is evaluated on a larger subset ($n = 401$, $K^\star = 10$) with budget $K \leq 15$, comparing random and low-coherence subsets. These choices reflect the intended regimes of the two methods rather than treating them as interchangeable solvers: BAL-A is used to study the minimum-flip attack and requires smaller attacked subsets, while BMP-A targets the budgeted $K$-sparse attack and can be evaluated on larger subsets. We also include a common feasible setting with $n = 50$ and $K^\star = 7$ for both methods to give a side-by-side comparison.

The SHP experiments have two downstream policy diagnostics. Attack-vs-groundtruth diagnostic retrains DPO on $\tilde{\mathcal{D}}(\hat{x})$ and $\tilde{\mathcal{D}}(x^\star)$ and compares $\pi_{\hat{\theta}}$ with $\pi_{\theta^\dagger}$, checking if the recovered flips reproduce the target-policy effect of the constructed attack. Clean-vs-attacked diagnostic retrains DPO on $\mathcal{D}$ and $\tilde{\mathcal{D}}(\hat{x})$ and compares $\pi_{\text{clean}}$ with $\pi_{\hat{\theta}}$, checking if the recovered flips actually mislead downstream policy away from clean policy. Full setup details are in Appendix H.

### 6.2. Results

#### 6.2.1. VALIDATE BAL-A THEORY ON SYNTHETIC $V$

Figure 1 shows the support TPR of BAL-A as the lattice penalty $M$ varies on a Gaussian synthetic dictionary. BAL-A recovers the true support almost perfectly for small $M$, but the recall drops quickly once $M$ becomes moderately large. The transition happens around the separation threshold $M_{\text{all sep}} \approx 0.68$ (vertical line) satisfying all of those in Theorem 4.3: below this value all the separation conditions hold and recovery is stable, while above it BAL-A increasingly miss the true flips. The binary-coefficient sufficient bound $M_0 \approx 1.69$ is overly conservative in this setting.

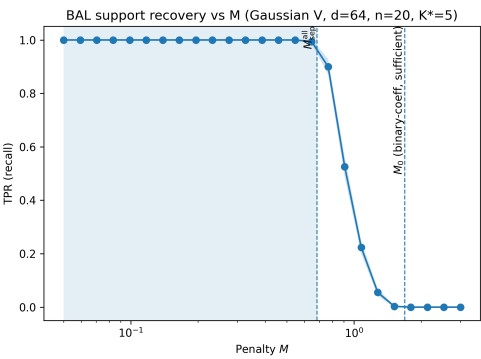

*Figure 1.* TPR of BAL-A on synthetic $V$ as a function of $M$.

### 6.2.2. VALIDATE BMP-A THEORY ON SYNTHETIC $V$

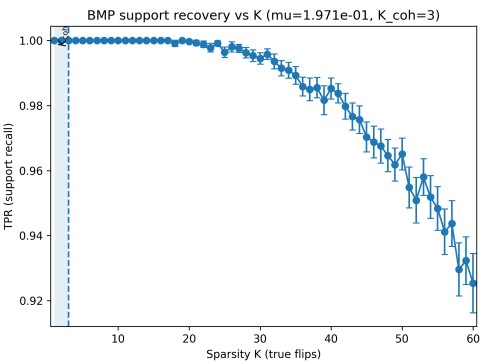

*Figure 2.* TPR of BMP-A on synthetic $V$ as a function of $K^\star$.

Figure 2 evaluates BMP-A on a fixed low-coherence synthetic dictionary while increasing the true number of flips $K^\star$. The constructed matrix has empirical coherence $\mu(V) \approx 0.197$ and the corresponding guarantee in Theorem 5.3 only covers up to $K_{\text{coh}} = 3$ (vertical line). Within this guaranteed region, BMP-A achieves perfect support recovery. Beyond $K_{\text{coh}}$, the sufficient condition no longer applies, yet BMP-A still maintains high TPR and degrades gradually as $K^\star$ grows, indicating that the coherence bound is conservative in this synthetic setting and that BMP-A remains effective outside the sufficient guarantee.

### 6.2.3. RESULTS ON REAL DATASET SHP

On SHP, the attack-vs-groundtruth results in Appendix I show that BAL-A and BMP-A can recover the constructed attack pattern and induce policies close to that trained on $\tilde{\mathcal{D}}(x^\star)$. BAL-A has perfect support recovery in the predicted $M$-s satisfying separation conditions, while BMP-A benefits from the low-coherence subset. The corresponding policy distances are $|\pi_{\theta\dagger} - \pi_{\hat\theta}| \approx 0.012$ in the SHP runs, indicating that training on $\tilde{\mathcal{D}}(\hat x)$ closely tracks training on $\tilde{\mathcal{D}}(x^\star)$.

Figure 3 gives the complementary clean-vs-attacked diag-

nostic on the common feasible SHP case. In this matched case, BAL-A and BMP-A recover the same constructed flip pattern exactly ($\text{TP} = 7, \text{FP} = 0, \text{FN} = 0$), so their attacked subsets coincide. This also allows a direct runtime check: BAL-A takes $0.6865$ seconds, while BMP-A takes $1.37 \times 10^{-4}$ seconds. We include this comparison only as a matched diagnostic, and the two methods are designed for different attack problems. The clean-vs-attacked distance

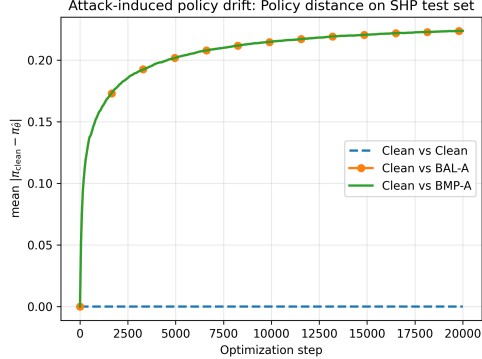

*Figure 3.* Clean-vs-attacked diagnostic. The curves overlap because BAL-A and BMP-A recover the same flip pattern.

$\|\pi_{\text{clean}} - \pi_{\hat\theta}\|_1$ stabilizes to around $0.224$, which is about 19 times the attack-vs-groundtruth distance of $0.012$. Thus the attack results do not merely recover the true flip pattern, they also mislead downstream DPO toward the target policy and away from the clean retraining baseline.

## 7. Conclusion

We have studied label flip preference attacks on offline RLHF, specifically targeting the log-linear DPO pipeline. We have illustrated the effect of flipping one preference label on the gradient of the DPO objective, and converted the attack problem to a binary sparse approximation problem in the per-sample gradient space. To solve it, we have proposed BAL-A algorithm, and provided sufficient conditions for binary coefficients and objective recovering. In practice, with budget $K$, we have also considered the $K$-flip attack problem and propose BMP-A, and provided a coherence-based recovery guarantee. We have also provided two impossibility conditions under which no $K$-flip attack can succeed. We have provided extensive experiments on both synthetic and real data to validate the theory.

Our goal is to understand the vulnerability of offline RLHF to adversarial preference poisoning, and to motivate robust algorithms that can defend against such attacks. As a first step, we limit our study to label flips in log-linear DPO with attacker access to white-box feature mapping $\psi$. These assumptions give an abstraction of general RLHF while leaving broader attack and defense settings for future work.

## Acknowledgements

The work of Chenye Yang and Lifeng Lai was supported in part by the National Science Foundation under grants CCF-2232907 and ECCS-2448268. The work of Weiyu Xu was supported by the National Science Foundation under grant ECCS-2133205.

## Impact Statement

This paper presents work whose goal is to advance the field of Machine Learning. There are many potential societal consequences of our work, none which we feel must be specifically highlighted here.

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

# A. Introduction: Additional Related Work

## A.1. Adversarial Attacks on Bandits

A line of work has studied adversarial attacks on bandit problems, including multi-armed bandits (MAB) (Jun et al., 2018; Liu & Shroff, 2019; Liu & Lai, 2020; Ma & Zhou, 2023; Yang et al., 2024), linear contextual bandits (Ma et al., 2018; Garcelon et al., 2020; Liu & Lai, 2023a), and dueling bandits (Gajane et al., 2015; Agarwal et al., 2021; Saha et al., 2021). These works have demonstrated that bandit algorithms are vulnerable to adversarial manipulation.

In the MAB setting, the learner takes an action by selecting one arm each round and observes its reward (Robbins, 1952; Auer et al., 1995; 2002). An adversary can manipulate the reward signals to mislead the learning algorithm into selecting a target arm more frequently (Jun et al., 2018; Liu & Shroff, 2019; Ma & Zhou, 2023), or corrupt the action signals to influence the learning process (Liu & Lai, 2020). Most of these existing work focused on the stationary random rewards setting, in which the distribution of reward of each arm does not change over time. Some (Ma & Zhou, 2023) studied the adversarial setting, in which the reward given by the environment can be arbitrarily chosen (Auer et al., 2002; Zimmert & Seldin, 2019). Other work (Yang et al., 2024) considered non-stationary stochastic bandits, in which the reward distribution of each arm can change over time but with some constraints on the total amount of changes (Garivier & Moulines, 2011; Besbes et al., 2014; Auer et al., 2019; Besbes et al., 2019; Cheung et al., 2022).

In linear contextual bandits, the expected reward is modeled as a linear function of the context features (Auer, 2002; Abe et al., 2003; Li et al., 2010). Existing works on adversarial attacks against linear contextual bandits focus on the reward poisoning (Ma et al., 2018; Garcelon et al., 2020), where the attacker manipulates the reward signal; context poisoning (Garcelon et al., 2020), where the adversary modifies the context observed without changing the reward associated with the context; and action poisoning (Liu & Lai, 2023a), where the adversary changes the action selected by the learner.

In dueling bandits, the learner selects a pair of arms each round and observes relative feedback indicating which arm is preferred (Yue et al., 2012; Zoghi et al., 2014; Sui et al., 2018). Prior work studies stochastic dueling bandits with adversarial corruption, where an attacker can corrupt pairwise outcomes (Agarwal et al., 2021). Other works consider fully adversarial preference models, which can be viewed as an extreme form of corruption (Gajane et al., 2015; Saha et al., 2021).

## A.2. Adversarial Attacks on Standard Reinforcement Learning

Adversarial attacks on standard reinforcement learning (RL) have been widely studied under different poisoning models, where the attacker modifies the learning signals to mislead the learned policy. Common corruptions include reward poisoning (manipulating reward or cost signals) (Huang & Zhu, 2019; Zhang et al., 2020; Rakhsha et al., 2020; 2021; Banihashem et al., 2022; Xu et al., 2023; Li et al., 2025), action poisoning (manipulating the agent's action before it is applied) (Liu & Lai, 2021), and more general environment poisoning such as manipulating transition dynamics (Rakhsha et al., 2020; Xu et al., 2021). Poisoning has also been studied in batch setting (Ma et al., 2019), and in dynamics-agnostic settings where attacker must learn to attack from interaction (Sun et al., 2021; Liu & Lai, 2021; Rakhsha et al., 2021; Xu et al., 2023; Li et al., 2025).

Beyond single-agent RL, adversarial attacks have been considered in more complex settings, including multi-agent RL where the attacker can corrupt rewards and/or actions (Liu & Lai, 2023b; Wu et al., 2023), and federated RL where poisoning can exploit both the RL dynamics and the federated aggregation mechanism (Ma et al., 2024). There is also work that characterizes limits of reward-only or action-only poisoning in episodic RL (Rangi et al., 2022), as well as work that studies optimal attack in attacker's Markov decision process (McMahan et al., 2024). In addition, many attacks are studied specifically in deep RL, where policies are neural networks and the attacker perturbs high-dimensional observations to induce harmful actions at decision time (Huang et al., 2017; Kos & Song, 2017; Lin et al., 2017; Mandlekar et al., 2017; Pattanaik et al., 2018; Gleave et al., 2020; Sun et al., 2020; Pan et al., 2022).

## A.3. Adversarial Attacks on Reinforcement Learning from Human Feedback

RLHF relies on human preference data as the training source, which is inherently different from bandit or standard RL settings that use numerical rewards. Recent works have shown that RLHF faces substantial security issues related to the reliability of preference data, i.e., when there are adversarial attacks on preferences or random noises in human preferences (Casper et al., 2023). In offline RLHF (Xiong et al., 2024), where the learner trains on a pre-collected preference dataset (Christiano et al., 2017; Ouyang et al., 2022; Bai et al., 2022; Zhu et al., 2023; Rafailov et al., 2023), preference data

can be manipulated by adversaries to mislead training and induce targeted behaviors (Rando & Tramèr, 2024; Baumgärtner et al., 2024; Wu et al., 2025a; Pathmanathan et al., 2025; Nika et al., 2025; Fu et al., 2025). Most existing studies demonstrate this vulnerability from data injection attacks and label flip attacks experimentally on large-scale models (Rando & Tramèr, 2024; Baumgärtner et al., 2024; Wu et al., 2025a; Pathmanathan et al., 2025; Fu et al., 2025), while (Nika et al., 2025) provides a comprehensive theoretical analysis of data injection attacks and shows that a large appended poisoned preference set may be needed to mislead offline RLHF. In online RLHF, where the agent learns from real-time interactions with human feedbacks (Chen et al., 2022; Saha et al., 2023; Wang et al., 2023; Zhan et al., 2024; Wu & Sun, 2024) adversaries also attack the preference to manipulate the learning process (Yang et al., 2025).

Besides adversarial poisoning, the preference data is also inherently noisy due to annotation difficulty, limited expert availability, and systematic differences across annotators (Casper et al., 2023; Yan et al., 2024). Such noise can lead to hard-to-predict behaviors and potential biases in aligned model outputs. Accordingly, a line of work studies robustness of offline RLHF under noisy or partially corrupted preferences (Yan et al., 2024; Cheng et al., 2024; Chowdhury et al., 2024; Mandal et al., 2025; Wu et al., 2025b). These works typically focus on random corruption models or random noise rather than a targeted adversarial attacker that strategically selects which comparisons to corrupt.

Compared with attacks on standard RL and bandits, attacks on RLHF face an additional challenge: the feedback is discrete preference data, and its effect on the learned policy is less direct and harder to predict. Moreover, in label flip attack problem for offline RLHF, the attacker is restricted to manipulating a fixed preference dataset, rather than freely manipulating rewards or transitions or generating new comparison pairs. Our work focuses on this practically plausible label flip threat model and provides a theoretical analysis for DPO under the log-linear policy class.

## B. Problem Formulation: Offline RLHF Pipeline and DPO Derivation (Rafailov et al., 2023)

This appendix provides a short summary that connects the standard offline RLHF pipeline to the DPO objective used in the main paper. We note that these derivations were done in (Rafailov et al., 2023) with a slightly different notation. We include them here for readers' convenience. The subsequent attack formulation and guarantees do not depend on the intermediate reward-modeling step.

RLHF frameworks usually combine three interconnected processes: feedback collection, reward modeling, and policy optimization (Christiano et al., 2017; Casper et al., 2023). Once given two answers $a_1$ and $a_2$ for the same prompt $s$, the human labeler is asked to compare the two answers and give their preference, denoted as $a_w \succ a_l \mid s$ where $a_w$ is the preferred answer and $a_l$ is the dispreferred answer in the pair $(a_1, a_2)$. Note that to match the notation in (Rafailov et al., 2023), this appendix writes each comparison as $(s, a_w, a_l)$, so the label is implicit. This is equivalent to the main-text notation $(s, a, a', o)$ with $o \in \{+1, -1\}$ by setting

$$(a_w, a_l) = \begin{cases} (a, a'), & o = +1, \\ (a', a), & o = -1. \end{cases}$$

Equivalently, given $(s, a_w, a_l)$ one can take $(s, a, a', o) = (s, a_w, a_l, +1)$.

We assume that there is some latent reward model $r^*(a, s)$ for the prompt $s$ and the answer $a$, which can not be observed by anyone. Bradley-Terry model (Bradley & Terry, 1952) describes the human preference distribution $p^*$ for pairwise comparisons:

$$p^*(a_1 \succ a_2 \mid s) = \frac{\exp(r^*(s, a_1))}{\exp(r^*(s, a_1)) + \exp(r^*(s, a_2))}.$$

In the offline RLHF setting, the agent has access to a static dataset of comparisons $\mathcal{D} = \{s^{(i)}, a_w^{(i)}, a_l^{(i)}\}_{i=1}^N$ sampled from $p^*$. Using the dataset $\mathcal{D}$, the agent can learn a reward model $r_\phi(s, a)$ parameterized by $\phi$ via maximum likelihood estimation (MLE). The negative log-likelihood loss function is defined as:

$$L_{\text{RLHF}}(r_\phi, \mathcal{D}) = -\mathbb{E}_{(s, a_w, a_l) \sim \mathcal{D}} \left[ \log \sigma(r_\phi(s, a_w) - r_\phi(s, a_l)) \right],$$

where $\sigma(x) = \frac{1}{1+\exp(-x)}$ is the sigmoid function for the Bradley-Terry model. Then a RL algorithm is used to optimize a policy $\pi_\theta$ parameterized by $\theta$ with the learned reward model $r_\phi(s, a)$:

$$\max_{\pi_\theta} \mathbb{E}_{s \sim \mathcal{D}, \, a \sim \pi_\theta(a|s)} \left[ r_\phi(s, a) - \beta \, \mathbb{D}_{\text{KL}} \left[ \pi_\theta(a \mid s) \, \| \, \pi_{\text{ref}}(a \mid s) \right] \right],$$

where $\beta$ is a parameter controlling the trade-off between learning the human preferences and following the reference policy $\pi_{\text{ref}}$.

To avoid the computational cost of policy optimization by reinforcement learning in large-scale problems, DPO (Rafailov et al., 2023) is proposed to directly optimize the policy $\pi_\theta$ with the preference data $(s, a_w, a_l)$. Solving the policy optimization problem leads to the following optimal solution:

$$\pi_r(a \mid s) = \frac{1}{Z(s)} \pi_{\text{ref}}(a \mid s) \exp\left(\frac{1}{\beta} r(s, a)\right),$$

where $Z(s) = \sum_a \pi_{\text{ref}}(a \mid s) \exp\left(\frac{1}{\beta} r(s, a)\right)$ is the partition function. Rearranging the above equation leads to:

$$r(s, a) = \beta \log \frac{\pi_r(a \mid s)}{\pi_{\text{ref}}(a \mid s)} + \beta \log Z(s).$$

Applying this reparameterization to the ground-truth reward function $r^*$ and corresponding optimal policy $\pi^*$, and the Bradley-Terry model, we have:

$$p^*(a_1 \succ a_2 \mid s) = \frac{1}{1 + \exp\left(\beta \log \frac{\pi^*(a_2|s)}{\pi_{\text{ref}}(a_2|s)} - \beta \log \frac{\pi^*(a_1|s)}{\pi_{\text{ref}}(a_1|s)}\right)}.$$

In this way, we model the human preference probability as a function of the optimal policy $\pi^*$ rather than the reward function $r^*$. Therefore, the loss function in MLE for parameterized policy $\pi_\theta$ can be rewritten as:

$$L_{\text{DPO}}(\pi_\theta; \pi_{\text{ref}}) = -\mathbb{E}_{(s, a_w, a_l) \sim \mathcal{D}}\left[\log \sigma\left(\beta \log \frac{\pi_\theta(a_w \mid s)}{\pi_{\text{ref}}(a_w \mid s)} - \beta \log \frac{\pi_\theta(a_l \mid s)}{\pi_{\text{ref}}(a_l \mid s)}\right)\right].$$

Solving this estimation problem leads to the desired policy $\pi_\theta$.

## C. Label Flip Attack: Extension Identities for General DPO and RLHF

This appendix records two natural extensions of Theorem 3.1.

### C.1. Extension Beyond Log-linear DPO

First, we consider DPO with differentiable policy classes beyond the log-linear class used in the main text, such as neural policies. In the log-linear case, the difference of the log-policy gradient reduces to a fixed feature difference, which makes each label flip a parameter-independent atom in the attack dictionary. Beyond this setting, the same algebra still gives an exact flip-induced gradient shift, but the atom is a policy-gradient difference and typically varies with $\theta$. The following proposition makes this distinction explicit: it defines a local dictionary at a target parameter $\theta^\dagger$, but does not by itself provide the fixed global dictionary used in the main attack reduction.

**Proposition C.1** (Flip-induced gradient shift for DPO with general differentiable policy classes). *Consider any differentiable policy class $\{\pi_\theta\}$ and a fixed reference policy $\mu$ that does not depend on $\theta$. For a comparison $(s_i, a_i, a'_i, o_i)$ with $o_i \in \{+1, -1\}$, define*

$$q_i(\theta) := \log \frac{\pi_\theta(a_i \mid s_i)}{\mu(a_i \mid s_i)} - \log \frac{\pi_\theta(a'_i \mid s_i)}{\mu(a'_i \mid s_i)}$$

*and the per-sample DPO loss $\ell_i(\theta; o_i) := -\log \sigma(o_i \beta q_i(\theta))$. If the label is flipped from $o_i$ to $-o_i$, then*

$$\nabla_\theta \ell_i(\theta; -o_i) - \nabla_\theta \ell_i(\theta; o_i) = o_i \beta \left(\nabla_\theta \log \pi_\theta(a_i \mid s_i) - \nabla_\theta \log \pi_\theta(a'_i \mid s_i)\right).$$

*Thus, the flip-induced gradient shift is exact but typically depends on $\theta$. At a fixed target $\theta^\dagger$, one may define local atoms*

$$v_i(\theta^\dagger) := o_i \beta \left(\nabla_\theta \log \pi_\theta(a_i \mid s_i) - \nabla_\theta \log \pi_\theta(a'_i \mid s_i)\right)\Big|_{\theta = \theta^\dagger}.$$

*If $V_{\theta^\dagger} x = -g^\dagger$, where $V_{\theta^\dagger} := [v_1(\theta^\dagger), \ldots, v_n(\theta^\dagger)]$ and $g^\dagger := \nabla_\theta L_{\text{DPO}}(\theta^\dagger; \mathcal{D})$, then $\theta^\dagger$ is a stationary point of the attacked DPO objective.*

**Challenges for extending to DPO with general policy classes.** The proposition shows that the flip-induced shift still has an exact first-order form, but extending the full log-linear theory to a general policy class faces three additional challenges. First, constructing $V_{\theta^{\dagger}}$ requires per-example policy-gradient computations, such as per-sample backpropagation through a neural policy, rather than simply forming fixed feature differences. Second, because the atoms $v_i(\theta)$ depend on the parameter value, the equation $V_{\theta^{\dagger}}x = -g^{\dagger}$ is only a local first-order condition at $\theta^{\dagger}$. For nonconvex DPO objectives, it does not by itself imply that $\theta^{\dagger}$ is a global optimizer of the attacked objective. Third, even a small gradient residual at $\theta^{\dagger}$ does not automatically yield the parameter-space and policy-space guarantees in Lemma 3.2 without strong convexity.

*Proof of Proposition C.1.* Since the reference policy $\mu$ is fixed, $\nabla_{\theta}q_i(\theta) = \nabla_{\theta}\log\pi_{\theta}(a_i \mid s_i) - \nabla_{\theta}\log\pi_{\theta}(a_i' \mid s_i)$. Differentiating $\ell_i(\theta; o_i) = -\log\sigma(o_i\beta q_i(\theta))$ gives

$$\nabla_{\theta}\ell_i(\theta; o_i) = -o_i\beta\,\sigma(-o_i\beta q_i(\theta))\,\nabla_{\theta}q_i(\theta).$$

After flipping the label,

$$\nabla_{\theta}\ell_i(\theta; -o_i) = o_i\beta\,\sigma(o_i\beta q_i(\theta))\,\nabla_{\theta}q_i(\theta).$$

Subtracting the two gradients and using $\sigma(t) + \sigma(-t) = 1$ yields

$$\nabla_{\theta}\ell_i(\theta; -o_i) - \nabla_{\theta}\ell_i(\theta; o_i) = o_i\beta\,\nabla_{\theta}q_i(\theta),$$

which proves the identity. The local stationarity claim follows by summing the identities over flipped indices: at $\theta^{\dagger}$, the attacked gradient is $\nabla_{\theta}L_{\mathrm{DPO}}(\theta^{\dagger}; \mathcal{D}) + V_{\theta^{\dagger}}x = g^{\dagger} + V_{\theta^{\dagger}}x$, which is zero if $V_{\theta^{\dagger}}x = -g^{\dagger}$. $\qquad\square$

## C.2. Extension to RLHF Reward Modeling

Next, we consider the standard two-stage RLHF pipeline with a separately learned reward model. Unlike DPO, where the preference loss directly updates the policy parameter, the first stage fits a reward parameter $\phi$ from the same type of pairwise comparison data. The following proposition shows that label flips induce an analogous gradient shift in the reward-modeling objective. This identity should be interpreted in reward-parameter space; its implication for the final policy depends on the subsequent policy-optimization stage.

**Proposition C.2** (Flip-induced gradient shift in the RLHF reward-modeling phase). *Consider the Bradley–Terry reward-modeling loss for a comparison $(s_i, a_i, a_i', o_i)$,*

$$\ell_i^{\mathrm{RM}}(\phi; o_i) := -\log\sigma\left(o_i\left(r_{\phi}(s_i, a_i) - r_{\phi}(s_i, a_i')\right)\right),$$

*where $r_{\phi}$ is a differentiable reward model. If the label is flipped from $o_i$ to $-o_i$, then*

$$\nabla_{\phi}\ell_i^{\mathrm{RM}}(\phi; -o_i) - \nabla_{\phi}\ell_i^{\mathrm{RM}}(\phi; o_i) = o_i\left(\nabla_{\phi}r_{\phi}(s_i, a_i) - \nabla_{\phi}r_{\phi}(s_i, a_i')\right).$$

*In particular, if the reward is linear, $r_{\phi}(s, a) = \phi^{\top}\psi(s, a)$, then the shift is the parameter-independent vector*

$$o_i\left(\psi(s_i, a_i) - \psi(s_i, a_i')\right).$$

*Equivalently, in winner-loser notation $(s_i, a_w^i, a_l^i)$ with $o_i = +1$, the shift is $\psi(s_i, a_w^i) - \psi(s_i, a_l^i)$.*

**Challenges for extending to RLHF.** The main difficulty is the two-stage nature of RLHF. The proposition gives a gradient identity only for the reward-modeling objective. A poisoned reward estimate must then pass through a separate policy-optimization stage, and the map from reward error to the final policy may depend on the RL optimizer, KL regularization, policy class, and possible nonconvexity. Therefore, the parameter-independent shift in reward-modeling phase does not by itself yield similar guarantees for final-policy manipulation as in log-linear DPO.

*Proof of Proposition C.2.* Let $h_i(\phi) := r_{\phi}(s_i, a_i) - r_{\phi}(s_i, a_i')$. Differentiating the reward-modeling loss gives

$$\nabla_{\phi}\ell_i^{\mathrm{RM}}(\phi; o_i) = -o_i\,\sigma(-o_ih_i(\phi))\,\nabla_{\phi}h_i(\phi).$$

After flipping the label,

$$\nabla_{\phi}\ell_i^{\mathrm{RM}}(\phi; -o_i) = o_i\,\sigma(o_ih_i(\phi))\,\nabla_{\phi}h_i(\phi).$$

Subtracting and using $\sigma(t) + \sigma(-t) = 1$ gives

$$\nabla_{\phi}\ell_i^{\mathrm{RM}}(\phi; -o_i) - \nabla_{\phi}\ell_i^{\mathrm{RM}}(\phi; o_i) = o_i\,\nabla_{\phi}h_i(\phi),$$

which is the stated general reward-modeling identity. If $r_{\phi}(s, a) = \phi^{\top}\psi(s, a)$, then $\nabla_{\phi}h_i(\phi) = \psi(s_i, a_i) - \psi(s_i, a_i')$, independent of $\phi$. $\qquad\square$

## D. BAL-A: Key Mismatches of the Standard Lattice Relaxation

This standard lattice approach has two key mismatches (binary coefficients, and minimum-flip objective) with our flip model, which can be further categorized into three detailed mismatches:

1. **Objective mismatch (minimum residual vs. minimum flips).** The CVP-style formulation focuses on minimizing the residual $\min_{x \in \{0,1\}^n} \|Vx + g^\dagger\|_2$, whereas our flip attack is a *minimum-cardinality* problem: $\min_{x \in \{0,1\}^n} \mathbf{1}^\top x$. Even when an exact attack is feasible (so the residual is zero), residual minimization alone does not distinguish among multiple feasible solutions and therefore does not, in general, recover the minimum-flip pattern.

2. **Unbounded integer coefficients.** In (9), the coefficients $z_i$ can be arbitrary integers, corresponding to "using" the same flip-effect $v_i$ multiple times. In our setting, each data point can be flipped at most once, so the admissible coefficients must lie in $\{0, 1\}$. Large integer coefficients are not meaningful as physical flip patterns.

3. **No guaranteed truncation.** A common workaround is to truncate an integer solution to the binary hypercube, e.g., $x_i = \mathbf{1}\{z_i \geq 1\}$. This nonlinear projection can destroy the relationship between $\|Vz + g^\dagger\|_2$ and $\|Vx + g^\dagger\|_2$, and there is no guarantee that a near-optimal integer solution leads to a near-optimal binary solution.

## E. BAL-A: Full Binary-Aware Lattice Attack Algorithm

The Binary-Aware Lattice Attack (BAL-A) algorithm is summarized in Algorithm 1.

Here we provide more details on each step of BAL-A:

1. **Construct the binary-aware lattice basis and target.**

    (a) Form the extended basis
    $$B_e = [b_1, \ldots, b_n] \in \mathbb{R}^{(d+n) \times n}, \quad b_i := \begin{pmatrix} v_i \\ M e_i \end{pmatrix},$$
    where $i = 1, \ldots, n$ and $e_i$ is the $i$-th standard basis vector in $\mathbb{R}^n$.

    (b) Define the target vector
    $$t := \begin{pmatrix} -g^\dagger \\ 0 \end{pmatrix} \in \mathbb{R}^{d+n}.$$
    Minimizing $\|B_e x - t\|_2$ over $x \in \mathbb{Z}^n$ is equivalent to minimizing $\|Vx + g^\dagger\|_2^2 + M^2 \|x\|_2^2$ in the binary-aware objective.

2. **Apply LLL basis reduction (Lenstra et al., 1982) to basis $B_e$.**

    (a) Fix a parameter $\delta \in (1/4, 1)$, e.g. $\delta = 3/4$.

    (b) Initialize the basis vectors $b_1, \ldots, b_n$ as the columns of $B_e$ and compute their Gram–Schmidt orthogonalization:
    $$b_i^*, \ \mu_{i,j}, \quad 1 \leq j < i \leq n,$$
    where
    $$b_i^* = b_i - \sum_{j<i} \mu_{i,j} b_j^*, \qquad \mu_{i,j} = \frac{\langle b_i, b_j^* \rangle}{\langle b_j^*, b_j^* \rangle}.$$

    (c) Set $k \leftarrow 2$.

    (d) While $k \leq n$, perform:

        i. *Size reduction:* for $j = k-1, k-2, \ldots, 1$, set
    $$m := \text{round}(\mu_{k,j}), \qquad b_k \leftarrow b_k - m \, b_j,$$
    and update the Gram–Schmidt data $(b_k^*, \mu_{k,\cdot})$.

        ii. *Lovász condition check:* if
    $$\delta \, \|b_{k-1}^*\|_2^2 \leq \|b_k^*\|_2^2 + \mu_{k,k-1}^2 \, \|b_{k-1}^*\|_2^2,$$
    then set $k \leftarrow k+1$ (the basis vectors $b_{k-1}, b_k$ are well-ordered). Otherwise (*swap step*):

- Swap the basis vectors: $(b_{k-1}, b_k) \leftarrow (b_k, b_{k-1})$.
- Recompute Gram–Schmidt data for indices $k - 1$ and $k$.
- Set $k \leftarrow \max\{k - 1, 2\}$.

(e) When the loop terminates, the vectors $b_1, \ldots, b_n$ form an LLL-reduced basis. Collect them into the reduced matrix $\tilde{B} \in \mathbb{R}^{(d+n) \times n}$.

(f) Keep track of the unimodular transformation $T \in \mathbb{Z}^{n \times n}$ such that $\tilde{B} = B_e T$, i.e., each reduced basis vector is an integer combination of the original ones.

3. **Run Babai's nearest-plane algorithm (Babai, 1986) on the reduced basis $\tilde{B}$.**

   (a) Compute a QR (or Gram–Schmidt) factorization of the reduced basis:

   $$\tilde{B} = QR,$$

   where $Q \in \mathbb{R}^{(d+n) \times n}$ has orthonormal columns and $R \in \mathbb{R}^{n \times n}$ is upper triangular.

   (b) Compute the coordinates of the target in the orthonormal basis:

   $$y := Q^\top t \in \mathbb{R}^n.$$

   (c) Initialize an integer coefficient vector $z \in \mathbb{Z}^n$ (e.g., $z = 0$).

   (d) For $i = n, n - 1, \ldots, 1$ (backwards):

   i. Compute the one-step residual

   $$r_i := y_i - \sum_{j=i+1}^{n} R_{ij} z_j.$$

   ii. Set

   $$z_i := \mathrm{round}\left(\frac{r_i}{R_{ii}}\right),$$

   where $\mathrm{round}(\cdot)$ denotes rounding to the nearest integer.

   (e) Map back to the original basis:

   $$x_{\mathrm{int}} := Tz \in \mathbb{Z}^n.$$

   By construction, $\tilde{B}z = B_e x_{\mathrm{int}}$, so the corresponding lattice vector is

   $$y(x_{\mathrm{int}}) = B_e x_{\mathrm{int}} - t = \begin{pmatrix} V x_{\mathrm{int}} + g^\dagger \\ M x_{\mathrm{int}} \end{pmatrix}.$$

4. **Truncation.**

   (a) For $M$ large enough, the integer vector $x_{\mathrm{int}}$ produced by the nearest-plane step satisfies $x_{\mathrm{int}} \in \{0, 1\}^n$. In that case, we directly take

   $$\hat{x} := x_{\mathrm{int}} \in \{0, 1\}^n$$

   as the binary flip pattern.

   (b) If in practice some entries of $x_{\mathrm{int}}$ fall outside $\{0, 1\}$, one may optionally project them back to $\{0, 1\}$, e.g.,

   $$\hat{x}_i := \min\{1, \max\{0, x_{\mathrm{int},i}\}\}, \quad i = 1, \ldots, n.$$

## F. BAL-A: Counterexamples for Surrogate Objective Mismatch

**Proposition F.1** (No universal $M$ yields minimum-flip recovery)**.** *Fix any $M > 0$. There exist an instance $(V, g^\dagger)$ for which the exact flip attack is feasible with optimal flip count $K^\star = 1$, i.e., $\exists\, x^\star \in \{0, 1\}^n$ with $\|x^\star\|_0 = 1$ and $V x^\star = -g^\dagger$. But the minimizer of the surrogate problem $\min_{x \in \{0,1\}^n} F_M(x)$ is infeasible and hence not the optimal attack pattern. Consequently, there is no universal choice of $M$ that guarantees $F_M$ recovers a feasible minimum-flip attack pattern for all instances.*

**Proposition F.2** (*M can enforce under-selection for any sparsity level*). *Fix any integer $K_0 \geq 2$ and any residual scale $r > 0$. There exists an instance $(V, g^\dagger)$ with optimal flip count $K^\star = K_0$, i.e., $\exists x^\star \in \{0, 1\}^n$ with $\|x^\star\|_0 = K_0$ and $Vx^\star = -g^\dagger$, such that for any $M$ satisfying $M^2 > r^2$ every minimizer $\hat{x}$ of $\min_{x \in \{0,1\}^n} F_M(x)$ satisfies $\|\hat{x}\|_0 \leq K_0 - 1$. In particular, minimizing $F_M$ does not recover the minimum-flip solution $x^\star$.*

Proposition F.1 shows that for any fixed $M$ there exist instances where the $F_M$ minimizer is infeasible even though a feasible exact solution exists. This rules out a universal $M$ that guarantees attack success via minimizing $F_M$. Proposition F.2 is conceptually different: it exhibits instances with any prescribed optimum sparsity $K^\star = K_0$ for which the $F_M$ strictly prefers an attack pattern with fewer flips. Together, these results clarify that $F_M$ mixes two goals: (i) promoting binary structure via regularization, (ii) recovering the minimum-flip attack pattern via residual minimization. These goals can conflict, so sufficient conditions tied to each need not admit a common choice of $M$ on a given instance.

## G. BAL-A: Worst-Case Performance of BAL-A

We cite the following corollary and definition on the worst-case performance of Babai's algorithm on LLL-reduced bases (Babai, 1986; Stephens-Davidowitz, 2016):

**Corollary G.1.** *For any $\delta \in (1/4, 1]$, Babai's algorithm with a $\delta$-LLL basis solves $\gamma$-CVP for $\gamma \leq \sqrt{n}\,(\delta - 1/4)^{-n/2}$. In particular, there is an efficient algorithm that solves $\gamma$-CVP for $\gamma = 2^{O(n)}$.*

**Definition G.2** ($\gamma$-CVP). *For any approximation factor $\gamma = \gamma(n) \geq 1$, the $\gamma$-Closest Vector Problem ($\gamma$-CVP) is defined as follows. The input is a basis for a lattice $\mathcal{L} \subseteq \mathbb{R}^n$ and a target vector $t \in \mathbb{R}^n$. The goal is to output a lattice vector $x \in \mathcal{L}$ with $\|x - t\| \leq \gamma \cdot \text{dist}(t, \mathcal{L})$.*

## H. Experiments and Results: Setup Details

This section collects implementation details omitted from the main text.

### H.1. Synthetic Dictionaries

We use synthetic $V$ to test the theory in a controlled way. For BMP-A, the coherence guarantee is stated for a fixed dictionary; therefore we fix a single low-coherence matrix $V \in \mathbb{R}^{200 \times 200}$ and sweep the sparsity level $K^\star$. We build $V$ by repeatedly resampling random Gaussian columns (normalized to unit norm) to reduce the maximum pairwise normalized correlation. The final constructed $V$ has empirical mutual coherence $\mu(V) \approx 0.1971$ and the maximum $K^\star$ satisfying the condition (15) is $K_{\text{coh}} = 3$. We then run 200 Monte Carlo trials per $K$ with $K \in \{1, \ldots, 60\}$. In each trial we sample a random support $S$ of size $K^\star$ as the ground-truth flips $x^\star$, set $-g^\dagger = \sum_{i \in S} v_i$, and run BMP-A for exactly $K^\star$ iterations to isolate whether the greedy rule selects the correct support as $K^\star$ increases.

For BAL-A, we study the dependence on the lattice penalty $M$ and compare to separation thresholds. Computing the separation threshold requires combinatorial search, so we keep $n$ small. We use a random Gaussian dictionary with unit-norm columns, with $V \in \mathbb{R}^{64 \times 20}$, $K^\star = 5$, and 200 Monte Carlo trials. The sufficient binary coefficient bound from (12) gives $M_0 \approx 1.68817$ with $B = 1$ and $R = 0$. The $M_{\text{all sep}}$ that satisfies all separation conditions in (13) is computed exactly as $M_{\text{all sep}} \approx 0.68287$. For each trial we sweep $M$ over 25 log-spaced values in $[0.05, 3.0]$; we apply LLL reduction $\delta = 0.75$ before Babai to improve numerical stability. With $n = 20$, we can enumerate all $k$-flip candidates for $k < K^\star$ and compute the separation threshold exactly, which is why we use this small-$n$ synthetic setting.

### H.2. Stanford Human Preferences Dataset

We also evaluate on a real preference dataset, Stanford Human Preferences (SHP) (Ethayarajh et al., 2022), by constructing $V$ from a log-linear DPO pipeline. We load 20,000 training pairs, of which each (history, response) pair is embedded with a frozen DistilBERT encoder (`distilbert-base-uncased`), using maximum sequence length 256 and the [CLS] embedding. For each preference pair we form the feature difference $\psi_i = \phi(\text{preferred}) - \phi(\text{non-preferred})$. We train a log-linear DPO policy for 3 epochs with learning rate $10^{-3}$, batch size 256, and $\beta = 1$. After training, we build the attack matrix $V$ such that each column $v_i$ is the per-example gradient change caused by flipping the $i$-th preference label.

BAL-A is expensive in the order of $n$, so on SHP we randomly pick the subset of $n = 50$ for training pairs attack $K^\star = 7$ among the subset, and run a penalty sweep over $M$ over 5 trials. The sufficient binary coefficient bound from (12) gives $M_0 \approx 6.71994$ with $B \approx 3.47103$ and $R = 0$. The $M_{\text{all sep}}$ that satisfies all separation conditions in (13) is computed

exactly as $M_{\text{all sep}} \approx 1.32166$. We use 25 log-spaced $M$ values in $[0.05, 10.0]$, BAL-A with LLL pre-reduction ($\delta = 0.75$) to attack. For each BAL-A trial on SHP, after fixing a SHP subset, we sample a support $S$ of size $K^\star$ within that subset, set $x^\star = \mathbf{1}_S$, and define the target by $t = Vx^\star = -g^\dagger$. BAL-A then attempts to recover $S$ from $(V, g^\dagger)$. Once we obtain the attack output $\hat{x}$, we evaluate support recovery against $x^\star$.

For BMP-A, we compare two subsets of the *same size* to isolate the effect of coherence: (i) a random subset of training pairs, and (ii) a low-coherence subset selected greedily by adding a column only if its maximum normalized inner product with the already-selected set is below a relaxed threshold. We use $K^\star = 10$ and a relaxed threshold larger than that in (15) so that the low-coherence subset is large enough to run many trials. Note that this threshold is not required to satisfy our sufficient condition; it is used only to reduce coherence in practice. On each subset we run 200 Monte Carlo trials: we sample $K^\star$ true flips uniformly within the subset, set $t = Vx^\star$, run BMP-A with a budget up to $K = K^\star + 5 = 15$, and use stopping tolerance $\varepsilon = 10^{-3}$. This is the same ground-truth construction as above: after fixing the subset and the support $S$, we define $x^\star = \mathbf{1}_S$ and then set the target by $t = Vx^\star = -g^\dagger$. We report support recovery and residual curves as a function of the budget, and we also report the empirical mutual coherence of each subset to quantify how the subset construction changes $V$.

The SHP experiments have two downstream policy diagnostics reporting parameter/policy distance. The attack-vs-groundtruth diagnostics compares DPO retrained on the BAL-A/BMP-A attacked subset $\tilde{\mathcal{D}}(\hat{x})$ with DPO retrained on the ground-truth attacked subset $\tilde{\mathcal{D}}(x^\star)$; this checks whether the recovered flips reproduce the target-policy effect of the constructed attack. The clean-vs-attacked diagnostic compares DPO retrained on the clean subset $\mathcal{D}$ with DPO retrained on $\tilde{\mathcal{D}}(\hat{x})$; this checks whether the recovered flips actually mislead downstream policy away from clean policy.

For attack-vs-groundtruth, we report $\|\hat{\theta} - \theta^\dagger\|_2$ and $\|\pi_{\hat{\theta}} - \pi_{\theta^\dagger}\|_1$ after retraining on $\tilde{\mathcal{D}}(\hat{x})$ and $\tilde{\mathcal{D}}(x^\star)$. For clean-vs-attacked, we use a matched feasible $n = 50$, $K^\star = 7$ SHP case with BAL-A ($M = 0.9$) and BMP-A ($K_{\max} = 7$). Both methods recover the sampled support exactly ($\text{TP} = 7, \text{FP} = 0, \text{FN} = 0$), so their attacked subsets coincide. BAL-A takes $0.6865$ seconds and BMP-A takes $1.37 \times 10^{-4}$ seconds. We retrain on $\mathcal{D}$ and $\tilde{\mathcal{D}}(\hat{x})$ for 20000 Adam steps with learning rate $10^{-3}$.

# I. Experiments and Results: SHP Diagnostics

This section reports the SHP experiments referenced in the main text. The results below give the support-recovery, residual, and attack-vs-groundtruth policy-distance diagnostics for BAL-A and BMP-A. Figure 3 in the main text gives the clean-vs-attacked diagnostic.

## I.1. BAL-A Performance on SHP

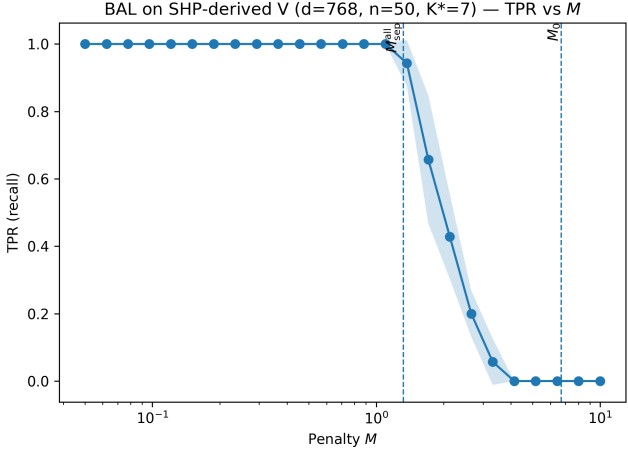

*Figure 4.* True positive rate of BAL-A on $V$ from SHP as a function of $M$.

Figure 4 reports TPR of BAL-A on an SHP-derived dictionary ($d = 768$, $n = 50$, $K^\star = 7$) as the penalty $M$ varies. Recovery is near-perfect for small $M$, and then degrades rapidly when $M$ exceeds the separation threshold $M_{\text{sep,all}}$, consistent with the theory. The explicit sufficient bound $M_0 \approx 6.72$ is conservative in this setting.

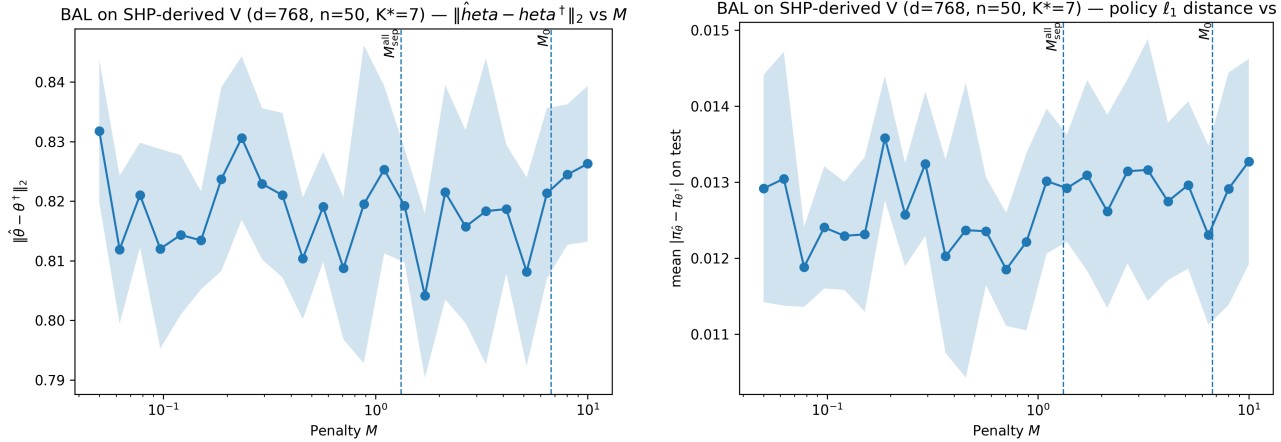

*Figure 5.* Attack-vs-groundtruth diagnostic for BAL-A: $\ell_2$ distance between learned parameters and $\ell_1$ distance between learned policies, comparing training on the BAL-A attacked subset $\tilde{\mathcal{D}}(\hat{x})$ versus training on the ground-truth attacked subset $\tilde{\mathcal{D}}(x^\star)$, as a function of $M$.

Figure 5 compares the learned model from the BAL-A attacked subset of SHP $\tilde{\mathcal{D}}(\hat{x})$ to the model learned from the ground-truth attacked subset of SHP $\tilde{\mathcal{D}}(x^\star)$, using $\|\hat{\theta} - \theta^\dagger\|_2$ in parameter space and $\|\pi_{\hat{\theta}} - \pi_{\theta^\dagger}\|_1$ in policy space. Across the full sweep of $M$, both distances change only mildly, even though support recovery in Fig. 4 collapses for large $M$. This weak dependence is expected in our SHP setting because the learned log-linear DPO model is trained on a limited subset ($n = 50$), so different flip sets can still yield similar fitted parameters/policies. These curves are the attack-vs-groundtruth policy diagnostic for BAL-A; together with TPR, they show when $\tilde{\mathcal{D}}(\hat{x})$ induces a policy close to that of $\tilde{\mathcal{D}}(x^\star)$. Because this experiment instantiates a log-linear DPO model on fixed preference pairs rather than a generative LLM, the most faithful downstream diagnostics here are parameter and policy distances after retraining, rather than qualitative free-form response examples.

### I.2. BMP-A Performance on SHP

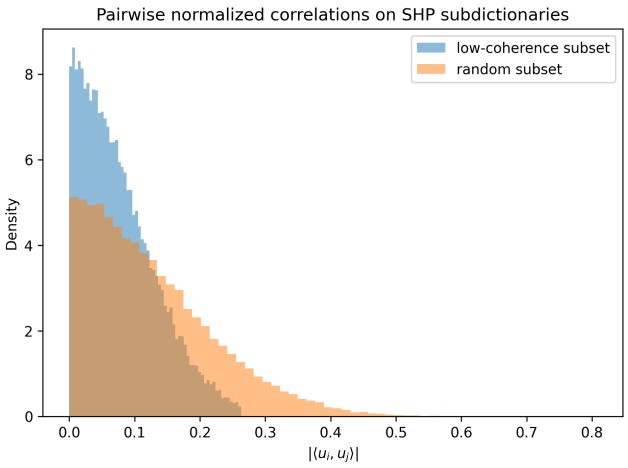

*Figure 6.* Histogram of pairwise normalized correlations for two subsets of SHP: a random subset and a low-coherence subset.

Figure 6 shows that the subset construction meaningfully changes the geometry of the SHP-derived dictionary. The random subset has a much heavier right tail of pairwise normalized correlations and a larger mutual coherence (here $\mu \approx 0.807$), while the low-coherence subset shifts the distribution left (here $\mu \approx 0.263$).

Figure 7 compares BMP-A on two SHP-derived dictionaries with the same size ($n = 401$) but different coherence: a random subset ($\mu = 0.807$, $K_{\text{coh}} = 1$) and a low-coherence subset ($\mu = 0.263$, $K_{\text{coh}} = 2$). We sample $K^\star = 10$ true flips

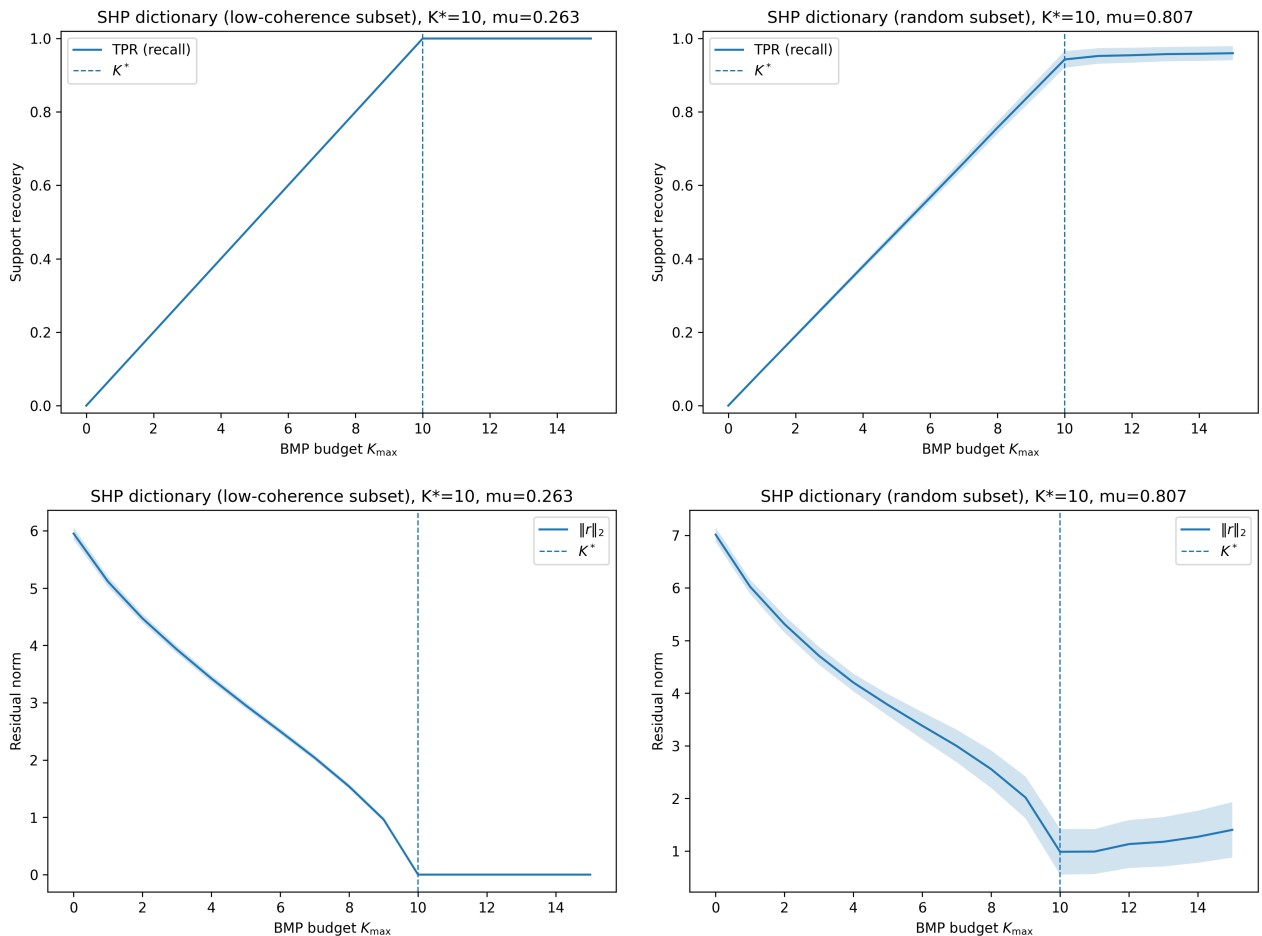

*Figure 7.* True positive rate and residual of BMP-A on $V$ from different subset of SHP as a function of budget $K$.

and set $y = V x^{\star}$, then run BMP-A with tolerance $\varepsilon = 10^{-3}$ up to budget $t_{\max} = 15$ with 200 trials. The low-coherence subset yields consistently higher TPR as the budget increases and drives the residual down faster, often reaching a near-zero residual around $K^{\star}$. In contrast, on the random subset BMP-A makes more incorrect selections as the budget grows, which limits support recovery and can increase the residual after later wrong updates. Overall, reducing coherence and increasing budget improves the greedy dynamics in practice.

Figure 8 evaluates $\|\hat{\theta} - \theta^{\dagger}\|_2$ and $\|\pi_{\hat{\theta}} - \pi_{\theta^{\dagger}}\|_1$ for the BMP-A. Over the sweep on budget $K$, both quantities change only modestly, even when the recovered support is imperfect. This is reasonable because the DPO model is trained on a larger but still limited subset ($n = 401$), so different flip sets can lead to similar fitted parameters and policies. These curves are the attack-vs-groundtruth policy diagnostic for BMP-A; together with TPR and residual reduction, they show whether $\tilde{\mathcal{D}}(\hat{x})$ produces a policy close to that of $\tilde{\mathcal{D}}(x^{\star})$.

# J. Proofs of Theorems

## J.1. Proof of Theorem 3.1

*Proof.* For the log-linear policy class, we have

$$\log \pi_\theta(a \mid s) = \psi(s, a)^{\top} \theta - \log \sum_b \exp(\psi(s, b)^{\top} \theta),$$

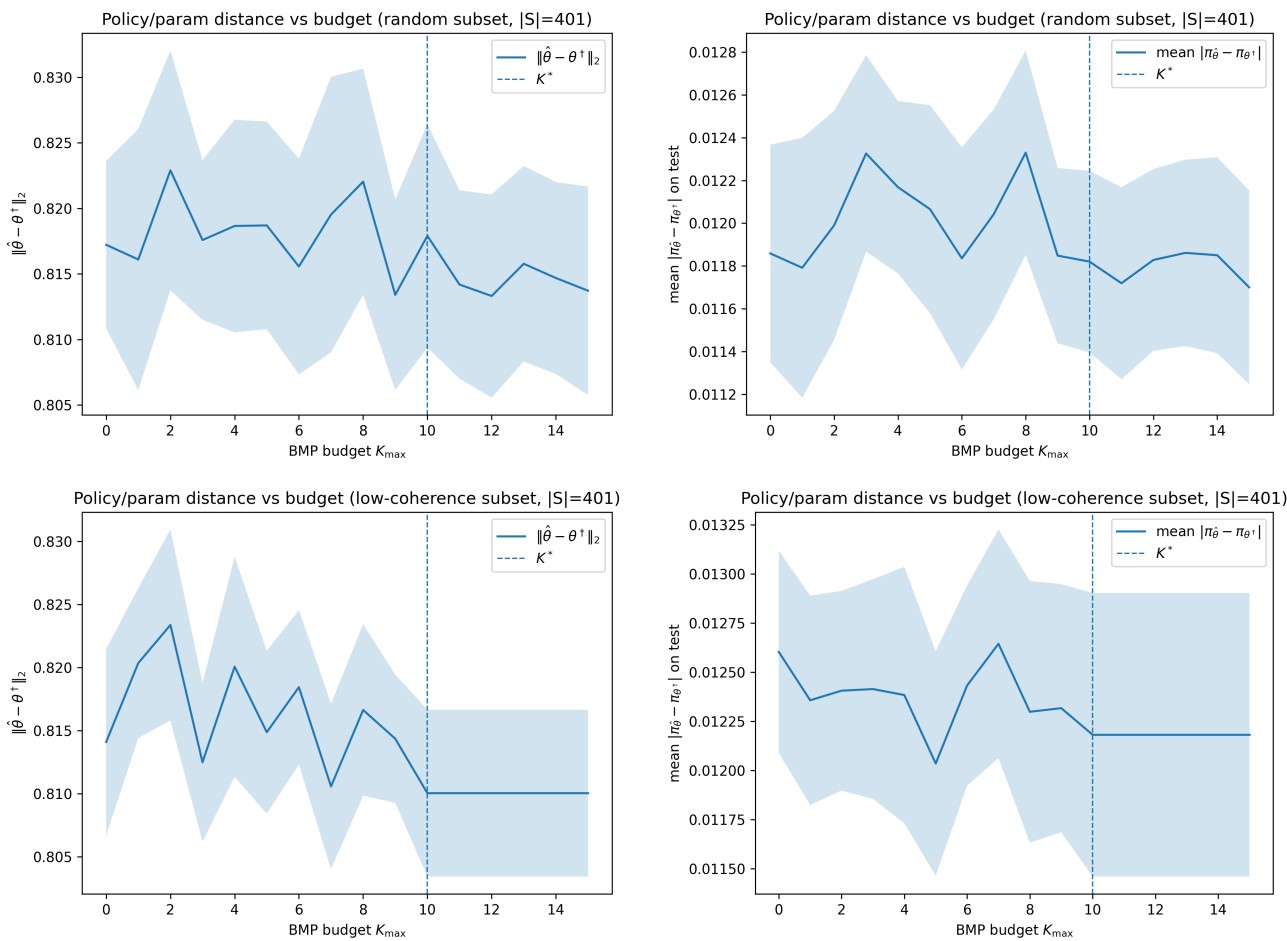

*Figure 8.* Attack-vs-groundtruth diagnostic for BMP-A: $\ell_2$ distance between learned parameters and $\ell_1$ distance between learned policies, comparing training on the BMP-A attacked $\tilde{\mathcal{D}}(\hat{x})$ versus training on the ground-truth attacked $\tilde{\mathcal{D}}(x^\star)$, as a function of budget $K$.

and similarly for $\mu = \pi_{\theta_\mu}$. Therefore,

$$
\log \frac{\pi_\theta(a_i \mid s_i)}{\mu(a_i \mid s_i)} - \log \frac{\pi_\theta(a_i' \mid s_i)}{\mu(a_i' \mid s_i)} = \psi(s_i, a_i)^\top \theta - \psi(s_i, a_i)^\top \theta_\mu - \psi(s_i, a_i')^\top \theta + \psi(s_i, a_i')^\top \theta_\mu = \Delta\psi_i^\top (\theta - \theta_\mu),
$$

where $\Delta\psi_i := \psi(s_i, a_i) - \psi(s_i, a_i') \in \mathbb{R}^d$.

Plugging into (1) gives per-sample loss

$$
\ell_i(\theta) = -\log \sigma\big(o_i \beta \Delta\psi_i^\top (\theta - \theta_\mu)\big) = \log\big(1 + \exp\big(-o_i \beta \Delta\psi_i^\top (\theta - \theta_\mu)\big)\big),
$$

since $-\log \sigma(t) = -\log\left(\frac{1}{1+e^{-t}}\right) = \log(1 + e^{-t})$. Differentiating yields the per-sample gradient

$$
g_i(\theta) = \nabla_\theta \ell_i(\theta) = -o_i \beta \, \sigma\big(-o_i \beta \Delta\psi_i^\top (\theta - \theta_\mu)\big) \Delta\psi_i.
$$

Now flip $o_i$ to $\tilde{o}_i = -o_i$. The new per-sample gradient is

$$
\tilde{g}_i(\theta) = -\tilde{o}_i \beta \, \sigma\big(-\tilde{o}_i \beta \Delta\psi_i^\top (\theta - \theta_\mu)\big) \Delta\psi_i = o_i \beta \, \sigma\big(o_i \beta \Delta\psi_i^\top (\theta - \theta_\mu)\big) \Delta\psi_i.
$$

Hence, the gradient shift is

$$
\Delta g_i(\theta) = \tilde{g}_i(\theta) - g_i(\theta) = o_i \beta \, \sigma\big(o_i \beta \, \Delta\psi_i^\top (\theta - \theta_\mu)\big) \Delta\psi_i + o_i \beta \, \sigma\big(-o_i \beta \, \Delta\psi_i^\top (\theta - \theta_\mu)\big) \Delta\psi_i.
$$

Using the sigmoid identity $\sigma(x) = 1 - \sigma(-x)$, we have

$$\Delta g_i(\theta) = o_i \beta \Delta \psi_i =: \Delta g_i.$$

Here we see that the gradient shift caused by flipping the label of one sample is a constant vector, independent of the parameter $\theta$. This proves Theorem 3.1. $\qquad\square$

## J.2. Proof of Lemma 3.2

*Proof.* Let $f(\theta) := L_{\mathrm{DPO}}(\theta; \tilde{\mathcal{D}})$ and $\theta^\star := \hat{\theta} = \arg\min_\theta f(\theta)$. Since $f$ is $m$-strongly convex and differentiable, see Lemma E.14 (Nika et al., 2025), its gradient is $m$-strongly monotone: for all $\theta$,

$$\langle \nabla f(\theta) - \nabla f(\theta^\star),\, \theta - \theta^\star \rangle \geq m\|\theta - \theta^\star\|_2^2.$$

Because $\theta^\star$ minimizes $f$, we have $\nabla f(\theta^\star) = 0$. Thus,

$$m\|\theta - \theta^\star\|_2^2 \leq \langle \nabla f(\theta),\, \theta - \theta^\star \rangle \leq \|\nabla f(\theta)\|_2 \, \|\theta - \theta^\star\|_2,$$

where the last inequality is Cauchy–Schwarz.

If $\theta \neq \theta^\star$, dividing both sides by $m\|\theta - \theta^\star\|_2$ yields $\|\theta - \theta^\star\|_2 \leq \|\nabla f(\theta)\|_2/m$, and the inequality is trivial when $\theta = \theta^\star$.

Taking $\theta = \theta^\dagger$ and using (4) together with (5) gives (6). Finally, Lemma E.5 of (Nika et al., 2025) shows that for log-linear policies, if $\|\hat{\theta}(x) - \theta^\dagger\|_2 \leq \epsilon/(2\sqrt{d})$, then $\|\pi_{\hat{\theta}(x)} - \pi_{\theta^\dagger}\|_1 \leq \epsilon$. Combining this with (6) yields the stated sufficient condition $\varepsilon \leq m\,\epsilon/(2\sqrt{d})$. $\qquad\square$

## J.3. Proof of Theorem 3.4

*Proof.* Recall that we assume that the feature embedding is uniformly bounded, e.g.,

$$\|\psi(s, a)\|_2 \leq 1.$$

Under this assumption, each feature difference satisfies

$$\|\Delta \psi_i\|_2 = \|\psi(s_i, a_i) - \psi(s_i, a_i')\|_2 \leq 2,$$

and hence each individual flip contribution is bounded as

$$\|o_i \beta \Delta \psi_i\|_2 = \beta\|\Delta \psi_i\|_2 \leq 2\beta.$$

**Exact attack.** In the exact attack setting, the constraint reads $g^\dagger = \sum_{i \in \mathcal{F}} o_i \beta\, \Delta \psi_i$. Taking norms and applying the triangle inequality, we obtain

$$\|g^\dagger\|_2 = \left\|\sum_{i \in \mathcal{F}} o_i \beta \Delta \psi_i\right\|_2 \leq \sum_{i \in \mathcal{F}} \|o_i \beta \Delta \psi_i\|_2 \leq \sum_{i \in \mathcal{F}} 2\beta = 2\beta|\mathcal{F}|. \tag{19}$$

Consequently,

$$|\mathcal{F}| \geq \frac{\|g^\dagger\|}{2\beta}. \tag{20}$$

**Approximate attack.** In the approximate case, we allow that $\left\|g^\dagger + \sum_{i \in \mathcal{F}} o_i \beta \Delta \psi_i\right\|_2 \leq \varepsilon$. Applying the reverse triangle inequality to the summation term, we have

$$\left\|\sum_{i \in \mathcal{F}} o_i \beta \Delta \psi_i\right\|_2 \geq \|g^\dagger\|_2 - \varepsilon.$$

On the other hand, as in (19), we also have

$$\left\|\sum_{i \in \mathcal{F}} o_i \beta \Delta \psi_i\right\|_2 \leq 2\beta|\mathcal{F}|.$$

Combining these inequalities yields

$$\|g^\dagger\|_2 - \varepsilon \leq 2\beta|\mathcal{F}|,$$

so that

$$|\mathcal{F}| \geq \frac{\|g^\dagger\|_2 - \varepsilon}{2\beta}. \tag{21}$$

$\square$

### J.4. Proof of Lemma 4.1

The proof is a local-improvement argument: if some coefficient were $\geq 2$, reducing it by one decreases the penalty by at least $3M^2$, while the residual term can increase by at most $O(B\|r\|) + O(B^2)$; choosing $M$ large ensures the penalty decrease dominates, contradicting optimality.

*Proof.* First, by feasibility of $x^\star$ and optimality of $x^{\mathrm{opt}}$,

$$\|y(x^{\mathrm{opt}})\|_2^2 \leq \|y(x^\star)\|_2^2 = \|Vx^\star + g^\dagger\|_2^2 + M^2\|x^\star\|_2^2 \leq R^2 + M^2K^\star. \tag{22}$$

Let

$$r := Vx^{\mathrm{opt}} + g^\dagger \in \mathbb{R}^d.$$

Then from (22) and $\|y(x^{\mathrm{opt}})\|_2^2 = \|r\|_2^2 + M^2\|x^{\mathrm{opt}}\|_2^2 \geq \|r\|_2^2$, we get

$$\|r\|_2 \leq \sqrt{R^2 + M^2K^\star}. \tag{23}$$

We now show that for $M$ large enough, $x^{\mathrm{opt}}$ cannot have any coordinate with magnitude at least 2. Suppose for contradiction that there exists an index $j$ with $|x_j^{\mathrm{opt}}| \geq 2$. Let $s := \mathrm{sign}(x_j^{\mathrm{opt}}) \in \{-1, +1\}$ and define the perturbed integer vector

$$\tilde{x} := x^{\mathrm{opt}} - se_j,$$

where $e_j$ is the $j$-th standard basis vector. Define the corresponding residual

$$\tilde{r} := V\tilde{x} + g^\dagger = r - sv_j.$$

Step 1 (bound the change in the top term). Compute

$$\|\tilde{r}\|_2^2 = \|r - sv_j\|_2^2 = \|r\|_2^2 - 2s\langle r, v_j\rangle + \|v_j\|_2^2,$$

hence

$$\|r\|_2^2 - \|\tilde{r}\|_2^2 = 2s\langle r, v_j\rangle - \|v_j\|_2^2.$$

Taking absolute values and using the triangle inequality (and $|s| = 1$),

$$\left|\|r\|_2^2 - \|\tilde{r}\|_2^2\right| = \left|2s\langle r, v_j\rangle - \|v_j\|_2^2\right| \leq 2|\langle r, v_j\rangle| + \|v_j\|_2^2.$$

By Cauchy–Schwarz,

$$|\langle r, v_j\rangle| \leq \|r\|_2 \|v_j\|_2,$$

so

$$\left|\|r\|_2^2 - \|\tilde{r}\|_2^2\right| \leq 2\|r\|_2\|v_j\|_2 + \|v_j\|_2^2.$$

Finally, using $\|v_j\|_2 \leq B$ yields

$$\left|\|r\|_2^2 - \|\tilde{r}\|_2^2\right| \leq 2B\|r\|_2 + B^2. \tag{24}$$

Using (23), we further obtain

$$\left|\|r\|_2^2 - \|\tilde{r}\|_2^2\right| \leq 2B\sqrt{R^2 + M^2K^\star} + B^2. \tag{25}$$

Step 2 (exact change in the bottom penalty). Since only the $j$-th coordinate changes,

$$\|x^{\mathrm{opt}}\|_2^2 - \|\tilde{x}\|_2^2 = (x_j^{\mathrm{opt}})^2 - (x_j^{\mathrm{opt}} - s)^2 = (x_j^{\mathrm{opt}})^2 - \left((x_j^{\mathrm{opt}})^2 - 2sx_j^{\mathrm{opt}} + 1\right) = 2sx_j^{\mathrm{opt}} - 1 = 2|x_j^{\mathrm{opt}}| - 1.$$

Because $|x_j^{\mathrm{opt}}| \geq 2$, we have $2|x_j^{\mathrm{opt}}| - 1 \geq 3$, hence

$$M^2\big(\|x^{\mathrm{opt}}\|_2^2 - \|\tilde{x}\|_2^2\big) \geq 3M^2. \tag{26}$$

Step 3 (combine and enforce positivity). By definition,

$$\|y(x^{\mathrm{opt}})\|_2^2 - \|y(\tilde{x})\|_2^2 = \big(\|r\|_2^2 - \|\tilde{r}\|_2^2\big) + M^2\big(\|x^{\mathrm{opt}}\|_2^2 - \|\tilde{x}\|_2^2\big).$$

Using (25) and (26),

$$\|y(x^{\mathrm{opt}})\|_2^2 - \|y(\tilde{x})\|_2^2 \geq 3M^2 - \Big(2B\sqrt{R^2 + M^2 K^\star} + B^2\Big). \tag{27}$$

If the right-hand side of (27) is strictly positive, then $\|y(\tilde{x})\|_2^2 < \|y(x^{\mathrm{opt}})\|_2^2$, contradicting optimality of $x^{\mathrm{opt}}$.

A sufficient (explicit) way to ensure positivity is to upper bound $\sqrt{R^2 + M^2 K^\star} \leq R + M\sqrt{K^\star}$, which yields the condition

$$3M^2 > 2B(R + M\sqrt{K^\star}) + B^2.$$

Equivalently,

$$3M^2 - 2B\sqrt{K^\star}\,M - (2BR + B^2) > 0,$$

which holds whenever $M$ exceeds the positive root of this quadratic, giving (12). Thus, for all such $M$, no coordinate of $x^{\mathrm{opt}}$ can satisfy $|x_j^{\mathrm{opt}}| \geq 2$. Therefore $x^{\mathrm{opt}} \in \{-1, 0, 1\}^n$. $\qquad\square$

## J.5. Proof of Theorem 4.2

*Proof.* By Lemma 4.1, every minimizer over $\mathbb{Z}^n$ satisfies $|x_i| \leq 1$ for all $i$ when $M \geq M_0$. Under the additional restriction $x_i \geq 0$, this implies $x_i \in \{0, 1\}$ for all $i$, hence $x^{\mathrm{opt}} \in \{0, 1\}^n$. The objective bound follows from feasibility of $x^\star$ and optimality of $x^{\mathrm{opt}}$: $\|y(x^{\mathrm{opt}})\|_2^2 \leq \|y(x^\star)\|_2^2 \leq R^2 + M^2 K^\star$. $\qquad\square$

## J.6. Proof of Theorem 4.3

*Proof.* Since $x^\star$ is feasible, $F_M(x^\star) = M^2 K^\star$. Consider any $x \in \{0, 1\}^n$. If $\mathbf{1}^\top x \geq K^\star$, then $F_M(x) \geq M^2 \mathbf{1}^\top x \geq M^2 K^\star = F_M(x^\star)$.

If $\mathbf{1}^\top x = k < K^\star$, then by definition of $\rho_k$,

$$F_M(x) \;\geq\; \rho_k^2 + M^2 k.$$

Under (13), $\rho_k^2 + M^2 k > M^2(K^\star - k) + M^2 k = M^2 K^\star = F_M(x^\star)$. Hence no $k$-flip vector with $k < K^\star$ can minimize $F_M$, and no vector with $k \geq K^\star$ can do better than $x^\star$ either. Therefore any minimizer must have $\mathbf{1}^\top x = K^\star$ and satisfy $Vx + g^\dagger = 0$, i.e., it is an optimal exact attack pattern. $\qquad\square$

## J.7. Proof of Proposition F.1

*Proof.* Fix any $M > 0$. Choose any dimension $d \geq 1$ and any $n \geq 1$. Let $g^\dagger \in \mathbb{R}^d$ be any nonzero vector with

$$\|g^\dagger\|_2 \;=\; \frac{M}{2} \;<\; M.$$

Construct $V \in \mathbb{R}^{d \times n}$ whose first column is

$$v_1 := -g^\dagger,$$

and whose remaining columns are arbitrary.

Then the exact flip attack is feasible with the binary vector $x^\star = e_1 \in \{0, 1\}^n$:

$$Vx^\star \;=\; v_1 \;=\; -g^\dagger,$$

so the optimal flip count is $K^\star = \|x^\star\|_0 = 1$.

Now consider the surrogate objective $F_M(x) = \|Vx + g^\dagger\|_2^2 + M^2 \mathbf{1}^\top x$ over $x \in \{0, 1\}^n$. For the zero vector,

$$F_M(0) \;=\; \|g^\dagger\|_2^2 \;=\; \frac{M^2}{4}.$$

For any $x \neq 0$ with $x \in \{0, 1\}^n$, we have $\mathbf{1}^\top x \geq 1$, hence

$$F_M(x) = \|Vx + g^\dagger\|_2^2 + M^2 \mathbf{1}^\top x \geq 0 + M^2 > \frac{M^2}{4} = F_M(0).$$

Therefore, the unique minimizer of $\min_{x \in \{0,1\}^n} F_M(x)$ is $\hat{x} = 0$.

Finally, since $g^\dagger \neq 0$, $\hat{x} = 0$ is infeasible for the exact attack constraint $Vx + g^\dagger = 0$ (it yields residual $\|g^\dagger\|_2 > 0$), whereas a feasible exact solution exists with one flip. This shows that for the fixed $M$ we constructed an instance where minimizing $F_M$ fails to recover a feasible minimum-flip exact attack. Since $M > 0$ was arbitrary, no universal choice of $M$ can guarantee minimum-flip recovery for all instances. $\qquad \square$

### J.8. Proof of Proposition F.2

*Proof.* Fix $K_0 \geq 2$ and $r > 0$. Choose any $d \geq K_0$ and any $n \geq K_0$. Construct a dictionary whose first $K_0$ columns are orthogonal vectors with equal norm $r$:

$$v_1, \ldots, v_{K_0} \in \mathbb{R}^d, \qquad \langle v_i, v_j \rangle = 0 \ (i \neq j), \qquad \|v_i\|_2 = r.$$

(The remaining columns, if any, are arbitrary.) Define

$$x^\star = \sum_{i=1}^{K_0} e_i \in \{0, 1\}^n, \qquad K^\star = \|x^\star\|_0 = K_0,$$

and set the noiseless target

$$-g^\dagger = Vx^\star = \sum_{i=1}^{K_0} v_i.$$

Then $x^\star$ is feasible and achieves zero residual, hence using $\|x^\star\|_2^2 = \|x^\star\|_0 = K_0$,

$$F_M(x^\star) = 0 + M^2 K_0.$$

Now consider the $(K_0 - 1)$-flip candidate $x' = \sum_{i=1}^{K_0-1} e_i$. Its residual is

$$Vx' - (-g^\dagger) = \sum_{i=1}^{K_0-1} v_i - \sum_{i=1}^{K_0} v_i = -v_{K_0},$$

so $\|Vx' - (-g^\dagger)\|_2^2 = \|v_{K_0}\|_2^2 = r^2$, and $\|x'\|_2^2 = K_0 - 1$. Therefore,

$$F_M(x') = r^2 + M^2(K_0 - 1).$$

Comparing $F_M(x')$ and $F_M(x^\star)$,

$$F_M(x') < F_M(x^\star) \quad \Longleftrightarrow \quad r^2 + M^2(K_0 - 1) < M^2 K_0 \quad \Longleftrightarrow \quad M^2 > r^2.$$

Hence for any $M$ satisfying $M^2 > r^2$, the $K_0$-flip solution $x^\star$ cannot minimize $F_M$, and every minimizer $\hat{x}$ must satisfy $\|\hat{x}\|_0 \leq K_0 - 1$. $\qquad \square$

### J.9. Proof of Theorem 5.3

*Proof.* We follow the proof of Theorem 1 in Appendix B of (Wen & Li, 2021) and only record the substitutions needed to match our non-normalized BMP-A.

**Notation substitutions.** Replace the unit-norm dictionary columns $A_i$ in (Wen & Li, 2021) by the normalized directions $u_i := v_i/\|v_i\|_2$, and denote $U := [u_1, \ldots, u_n]$. Replace the measurement model $y = Ax^\star + v$ by our target $y$ together with the residual noise $e := y - Vx^\star$ (so $\|e\|_2 \leq \varepsilon$ by Assumption 5.1). Replace their sparsity level $K$ by $K^\star$, their tolerance $\delta$ by $\varepsilon$, and their coherence $\mu(A) = \max_{i \neq j} |A_i^\top A_j|$ by $\mu(V) := \max_{i \neq j} |\langle u_i, u_j \rangle|$. Finally, introduce the column-norm range $b := \min_i \|v_i\|_2$ and $B := \max_i \|v_i\|_2$.

**Changes to Appendix B inequalities.** The greedy selection rule of BMP-A is

$$i_k \in \arg\max_{i \notin \Gamma_{k-1}} \frac{|\langle v_i, r^{k-1}\rangle|}{\|v_i\|_2} = \arg\max_{i \notin \Gamma_{k-1}} |\langle u_i, r^{k-1}\rangle|,$$

so (B1) in Appendix B of (Wen & Li, 2021) becomes

$$\|U_{\Omega \setminus \Gamma_{k-1}}^\top r^{k-1}\|_\infty > \|U_{\Omega^c}^\top r^{k-1}\|_\infty.$$

The residual decomposition (B2) is replaced by the exact identity implied by our update $r \leftarrow r - v_i$:

$$r^{k-1} = y - \sum_{i \in \Gamma_{k-1}} v_i = \sum_{i \in \Omega \setminus \Gamma_{k-1}} v_i + e = \sum_{i \in \Omega \setminus \Gamma_{k-1}} \|v_i\|_2 \, u_i + e.$$

With these substitutions, the derivation of (B3) is unchanged, except that every occurrence of $A^\top(\cdot)$ is replaced by $U^\top(\cdot)$ and $A_{\Omega \setminus \Gamma} x_{\Omega \setminus \Gamma}$ is replaced by $V_{\Omega \setminus \Gamma} \mathbf{1}$, since $x^\star$ is binary on the remaining support.

The two key bounds (B4) and (B5) in Appendix B of (Wen & Li, 2021) now read

$$\|U_{\Omega \setminus \Gamma}^\top V_{\Omega \setminus \Gamma} \mathbf{1}\|_\infty \geq b - (|\Omega \setminus \Gamma| - 1)\mu(V)B \quad \text{and} \quad \|U_{\Omega^c}^\top V_{\Omega \setminus \Gamma} \mathbf{1}\|_\infty \leq |\Omega \setminus \Gamma|\mu(V)B,$$

since $\langle u_i, v_i \rangle = \|v_i\|_2 \geq b$ and $|\langle u_i, v_j \rangle| = \|v_j\|_2 |\langle u_i, u_j \rangle| \leq B\mu(V)$ for $i \neq j$. The noise bound (B6) is unchanged because $\|u_i\|_2 = 1$:

$$\|U_{\Omega \setminus \Gamma}^\top e\|_\infty + \|U_{\Omega^c}^\top e\|_\infty \leq 2\|e\|_2 \leq 2\varepsilon.$$

Combining these bounds yields the same correlation gap as in Appendix B of (Wen & Li, 2021) provided that

$$b - (2K^\star - 1)\mu(V)B > 2\varepsilon > 0,$$

which is exactly (15)–(16). Therefore, BMP-A selects an index in $\Omega = \mathrm{supp}(x^\star)$ at each iteration, and after $K^\star$ iterations it recovers $\mathrm{supp}(x^\star)$.

Finally, once $\mathrm{supp}(\hat{x}) = \mathrm{supp}(x^\star)$, we have $r^{K^\star} = y - V\hat{x} = y - Vx^\star = e$, so $\|r^{K^\star}\|_2 \leq \varepsilon$ and the stopping rule triggers no later than iteration $K^\star$. □

### J.10. Proof of Theorem 5.4

*Proof.* We prove impossibility by upper bounding the largest gradient shift $\|Vx\|_2$ achievable with at most $K$ flips, and comparing it to the amount of shift needed to reach tolerance $\varepsilon$.

Suppose there exists a binary vector $x \in \{0,1\}^n$ with $\|x\|_0 \leq K$ such that $\|Vx + g^\dagger\|_2 \leq \varepsilon$. Let $r := Vx + g^\dagger$. By the reverse triangle inequality,

$$\|Vx\|_2 \geq \|g^\dagger\|_2 - \|r\|_2 \geq \|g^\dagger\|_2 - \varepsilon. \tag{28}$$

Thus any successful $K$-flip attack must produce a shift of size at least $\|g^\dagger\|_2 - \varepsilon$.

**Spectral norm impossibility** (17). Since $x$ is binary with at most $K$ ones, we have $\|x\|_2 = \sqrt{\|x\|_0} \leq \sqrt{K}$. Therefore,

$$\|Vx\|_2 \leq \|V\|_2 \|x\|_2 \leq \|V\|_2 \sqrt{K}.$$

Combining with (28) yields

$$\|g^\dagger\|_2 - \varepsilon \leq \|V\|_2 \sqrt{K},$$

which contradicts (17).

**Coherence impossibility** (18). Let $S := \mathrm{supp}(x)$ so that $|S| = \|x\|_0 \le K$ and $Vx = \sum_{i \in S} v_i$. Write $v_i = a_i u_i$ with $a_i = \|v_i\|_2$ and $\|u_i\|_2 = 1$. Expanding the squared norm and using $|\langle u_i, u_j \rangle| \le \mu(V)$ gives

$$
\begin{aligned}
\|Vx\|_2^2 &= \left\langle \sum_{i \in S} a_i u_i, \sum_{j \in S} a_j u_j \right\rangle = \sum_{i \in S} a_i^2 + \sum_{\substack{i,j \in S \\ i \ne j}} a_i a_j \langle u_i, u_j \rangle \\
&\le \sum_{i \in S} a_i^2 + \mu(V) \sum_{\substack{i,j \in S \\ i \ne j}} a_i a_j \le |S| B^2 + \mu(V) |S|(|S|-1) B^2 \le B^2 \Big( K + \mu(V) K(K-1) \Big).
\end{aligned}
\tag{29}
$$

Together with (28), this implies

$$
\left( \|g^\dagger\|_2 - \varepsilon \right)^2 \le B^2 \Big( K + \mu(V) K(K-1) \Big),
$$

which contradicts (18). This completes the proof. $\qquad\square$

