# OpenReview forum: "Efficient Preference Poisoning Attack on Offline RLHF"
_ICML.cc/2026/Conference — ICML 2026 regular_

### Official Review · Reviewer_cBSx · 2026-02-25

**Soundness:** 3
**Presentation:** 2
**Significance:** 3
**Originality:** 3
**Overall Recommendation:** 5
**Confidence:** 4

**Summary:**

This paper considers the problem of label-flipping attacks for DPO. Using a key insight derived in the paper that the gradient shifts by a parameter-invariant amount when labels flip, the authors formulate a well-defined attack optimization problem with the goal of enforcing a target policy $\pi^\dagger$. They first provide lower bounds on the flips necessary to enforce $\pi^\dagger$ onto a DPO learner. Then, they consider two variants of the problem, one with unrestricted budget, and one with budget constraints. For the first problem, they propose a relaxation of the original problem as a binary-aware lattice attack problem and propose the BAL-A algorithm with guarantees to minimize the residual and also enforce the binary condition on the flip vector. For the budget-constrained formulation, they propose the BMP-A algorithm which is supposed to solve an instantiation of a sparse approximation problem, corresponding to the attack problem. Finally, they validate their algorithms on synthetic and the Stanford Human Preferences data.

**Compliance With Llm Reviewing Policy:**

Affirmed.

**Final Justification:**

Authors have addressed my concerns.

**Key Questions For Authors:**

**Questions and Suggestions**

1. I would be interested in seeing a result that provides conditions when Assumption 3.3 holds, and when it breaks, depending on the structure of $\pi^\dagger$ and the clean dataset $\mathcal{D}$. For instance, I would expect that the closer $\pi^\dagger$ is to the optimal policy induced by $\mathcal{D}$, the higher the probability that Assumption 3.3 is satisfied.
2. You're using $\mu$ for both strong convexity factor and reference policy. It might be a bit confusing.
3. Theorem 3.4 suggests that, the lower the regularization parameter (assuming $\beta\in (0,1)$), the more flips are required for a successful attack. I would like to understand if there is an intuition behind this. On first glance, I would expect more attack budget to be used against a policy that tends to remain close to $\mu$, not less.
4. Theoretical results of Section 4 and 5 deserve more attention. There should be some corollaries of Theorem 5.4 for example. What does this mean for the budgeted attack in special cases? At least some remarks are in order.
5. What is the relationship between the considered setting and label-flipping attacks to an RLHF learner, say, with linear rewards and log-linear policies? Is it hard to extend such results there and if so, why? To me it seems like that should be the natural starting point. Have you considered such a problem? On first glance, it seems to me that your Theorem 3.1 should extend to the reward-learning phase of RLHF, right?

**Limitations:**

Authors have not discussed limitations in the main paper. They should be discussed. For example, loglinearity assumption, and the breadth of the experiments are two of the limiting factors.

**Strengths And Weaknesses:**

**Strengths**

1. I believe the problem is significant and relevant to the community. And the theoretical results seem sound and do advance the understanding of label-flipping attacks on DPO, from a theoretical standpoint.
2. The usage of the proposed algorithms designed to solve this attack problem seems original to me. Apart from that, the derived theory also sheds light onto particular structure of the label-flipping attack to DPO.

**Weaknesses**

1. The presentation could benefit from more insights/remarks/discussion on the derived results. I believe there are many consequences of the results that deserve the attention of the reader, which are not currently provided in the paper. Also, to me it seems that the pseudocodes of the proposed algorithms should be included in the main paper and there should also be a more intuitive insight (diagrams or sketches might help and would be highly recommended!) to better illustrate the lattice and sparse approximation formulations.
2. I was expecting more attention to be given to the real data experiments. But only a single paragraph is devoted to it. Apart from empirically illustrating that the algorithm is working properly (in the sense of finding the approximate solution that is designed to find), I don't see any other consequences.

---

> ### Author Rebuttal · Authors · 2026-03-30
>
> # Reply to Reviewer cBSx
>
> Thank you for constructive comments. Below we respond point-by-point and will incorporate these clarifications in the revision.
>
> ## Question 1
>
> Studying when Assumption 3.3 holds is valuable. In general, however, the attack problem is a binary sparse approximation problem, which is NP-hard, so a full feasibility check is difficult. This is why we impose Assumption 3.3, and then complement it with the lower bound in Theorem 3.4, and the impossibility condition in Theorem 5.4.
>
> That said, the reviewer’s intuition is partly correct: if $\pi^\dagger$ is closer to the clean optimum $\pi^\star$, then $|g^\dagger|$ is typically smaller (exactly zero if $\pi^\dagger=\pi^\star$), which generally makes the attack easier. But feasibility depends not only on $\pi^\dagger$, but also on the geometry of the dictionary $V$, i.e., whether $-g^\dagger$ can be well approximated by a sparse binary combination of its columns. So closeness of $\pi^\dagger$ to $\pi^\star$ is helpful but not by itself sufficient.
>
> ## Question 2
>
> We will change the strong-convexity notation in Lemma 3.2 and its proof from $\mu$ to $m$.
>
> ## Question 3
>
> In DPO, $\beta$ scales the log-likelihood ratio of $\pi_\theta$ against the reference $\mu$. But Theorem 3.4 is not about how close the policy is to $\mu$; it concerns how much one flipped label can change the gradient. In log-linear DPO, flip-induced gradient shift is $$o_i \beta \Delta \psi_i,$$
> so its norm is proportional to $\beta$. Thus, smaller $\beta$ makes each individual flip weaker, so more flips may be needed to cancel $g^\dagger$. This is why the lower bound in Theorem 3.4 scales like $1/\beta$.
>
> ## Question 4
>
> We discuss special cases of Theorem 5.4. When $K=1$, the theorem implies that no single-flip attack can succeed if $\|g^\dagger\|_2-\varepsilon > B$. When $B=1$ in the normalized case, the coherence condition becomes
> $$(\|g^\dagger\|_2-\varepsilon)^2 > K + \mu(V)K(K-1),$$
> which makes the roles of budget and coherence explicit. If columns are weak (small $B$) and diverse (small $\mu(V)$), even the best $K$ flips cannot create a large enough shift to cancel $g^\dagger$.
>
> In revision, we will add discussion after theorems to explain their practical implications, especially the roles of $M$, $K$, $\|g^\dagger\|_2$, and the geometry of $V$.
>
> ## Question 5
>
> An analogue of Theorem 3.1 indeed holds in the *reward-modeling phase* of RLHF with linear rewards.
>
> Consider B-T model with linear reward
> $$r_\phi(s,a)=\phi^\top \psi(s,a),$$
> and one preference pair $(s,a_w,a_l)$. Let
> $$\Delta \psi := \psi(s,a_w)-\psi(s,a_l),\qquad z_\phi := r_\phi(s,a_w)-r_\phi(s,a_l)=\phi^\top \Delta\psi.$$
> The per-sample reward-modeling loss is
> $$\ell_\phi(s,a_w,a_l)=-\log \sigma(z_\phi).$$
> Thus, the flip-induced gradient shift is
> $$\nabla_\phi \ell_\phi^{\mathrm{flip}}(s,a_w,a_l)-\nabla_\phi \ell_\phi(s,a_w,a_l)=\big(\sigma(z_\phi)+\sigma(-z_\phi)\big)\Delta\psi=\Delta\psi,$$
> which is again *parameter-independent*.
>
> However, the full two-stage RLHF is different from DPO. Even if the reward-learning phase admits the same identity, one must still analyze how the poisoned reward propagates through subsequent policy-optimization stage. This extra stage is one reason why we focus on DPO: it gives a direct objective over the policy parameter, allowing cleaner theory.
>
> For completeness, the per-sample gradient shift for *general DPO* is
> $$\beta\Big(\nabla_\theta \log \pi_\theta(a_w\mid s)-\nabla_\theta \log \pi_\theta(a_l\mid s)\Big),$$
> which is exact but typically $\theta$-dependent. The log-linear case is special because the difference reduces to parameter-independent Theorem 3.1.
>
> ## Weakness 1
>
> Due to page limit, we focused the main text on the formulation, methods, etc. In revision with one extra page, we will add pseudocode of attacks, more intuitive explanations, and some simple sketches.
>
> ## Weakness 2
>
> We presented the SHP experiments in Appendix H and I, which report the success of attack pattern recovery and downstream attack-vs-groundtruth DPO policy distance. We agree with the reviewer that other attack consequences, e.g., a clean-vs-attacked comparison on $\mathcal D$ and $\tilde{\mathcal D}(\hat x)$ by BAL-A/BMP-A, would make attack effects clearer.
>
> We run additional experiment to include this comparison, and use the same downstream metrics (parameter/policy distance). Detailed setup and results can be found in Reply to Weakness 2 of reviewer **mfhS**.
>
> Retraining on the attacked subset yields a *clear downstream deviation* from retraining on the clean subset: $| \pi_{\text{clean}} - \pi_{\hat{\theta}} | \approx 0.224$. By contrast, Appendix I reports $| \pi_{\theta^\dagger} - \pi_{\hat{\theta}} | \approx 0.012$, which is only about 5\% of the clean-policy gap. Thus the attacks move the learned policy much closer to the target than to the clean policy.
>
> ## Limitation
>
> In revision, we will add the discussion of limitations (details in Reply to Limitation of reviewer **DQfh**).

---

> > ### Author Rebuttal · Reviewer_cBSx · 2026-04-03
> >
> > I thank the authors for their rebuttal. My concerns and questions have been resolved. I encourage the authors to integrate all the provided answers (remarks, pseudocodes, experiments, and notation) in the final version of the paper, since, to me, they would definitely strengthen its presentation. I increase my score.

---

> > > ### Author Response · Authors · 2026-04-03
> > >
> > > We thank the reviewer again for your efforts in reviewing our paper and for your constructive comments. We are glad to hear that we have adequately addressed all your concerns and that the reviewer has raised the score. We will definitely integrate all the provided answers in the final version, which would greatly strengthen the presentation of the paper. We are very grateful for the reviewer's support.

---

### Official Review · Reviewer_DQfh · 2026-03-02

**Soundness:** 3
**Presentation:** 3
**Significance:** 3
**Originality:** 3
**Overall Recommendation:** 4
**Confidence:** 3

**Summary:**

The paper studied the label flip attack for the preference label in Reinforcement Learning from Human Feedback (RLHF) under offline setting. By transfering the problem to a structured binary sparse approximation problem by utilizing the property of gradients, the authors proposed two methods called Binary Aware Lattice Attack (BAL-A) and Binary Matching Pursuit Attack (BMP-A) for without budget for flip and with budget respectively. Finally, they also run some experiments to verify their theretical findings.

**Compliance With Llm Reviewing Policy:**

Affirmed.

**Final Justification:**

I keep my original score. The authors solved my concerns in rebuttal and I believe they will revise it in the next version.

**Key Questions For Authors:**

1. How good the theretical results? For example, the upper bound in Theorem 4.2, how to explain how good is it? Is there any lower bound to show its optimality or does it outperform than the bounds in previous work?

2. As you mentioned in problem formulation, the main objective of RLHF or DPO is to estimate $\theta$. In your work, what method you are using to estimate it. Under the label flip attack problem, what is the effect of the problem to estimate $\theta$?

3. The method part (algo 1 and algo 2) are based on previous work and modified from (Wen & Li, 2021). What the main novelty for your mehtod or technique?

**Limitations:**

yes

**Strengths And Weaknesses:**

Strengths:
1. The writing is clear, and the authors provided comprehensive information in aapendix such as related work and additional knowledge of DPO.

2. The authors described the logic of the technique for solving the problem in a decent way.

3. The paper derived solid theoretical results and ran some experiments to support the findings.

Weaknesses:

1. The method to solve the problem is mainly modified from previous work.

2. Please see some other concerns in the Questions part.

---

> ### Author Rebuttal · Authors · 2026-03-30
>
> # Reply to Reviewer DQfh
>
> Thank you for the careful review and constructive comments. Below we respond point-by-point and will incorporate these clarifications in the revision.
>
> ## Question 1
>
> The result corresponding to Theorem 4.2 is a *sufficient* guarantee, not a claim of tight optimality. Its role is to show that there exists a provable binary-enforcing method for the lattice embedding; we do not claim that the explicit threshold is sharp. In fact, our experiments already indicate that the bound is conservative, and we will say this explicitly in the revised discussion after the theorem. To the best of our knowledge, we are not aware of a directly comparable prior bound for this exact binary-aware lattice-embedding setting. Classical LLL/Babai results give generic approximation guarantees for lattice decoding, such as Babai (1986), but these do not analyze the same augmented Euclidean objective with an explicit binary-enforcing threshold as in Theorem 4.2. Therefore, we do not claim a direct optimality comparison against prior bounds. Theorem 4.3 and Theorem 5.3 similarly give sufficient conditions for aligning the minimum-flip objective and recovering the true attack pattern, but we do not claim that these conditions are necessary or tight.
>
> ## Question 2
>
> In the paper, the learner estimates $\hat\theta$ by minimizing the regularized DPO objective (second equation in Section 2.1). In the theory, $\hat\theta$ denotes the exact minimizer of that objective; in the experiments, we compute an approximate minimizer by standard gradient-based optimization of the same loss, e.g., Adam or SGD. Under label flipping, the optimality condition changes from $\nabla L_{\mathrm{DPO}}(\hat\theta;\mathcal D)=0$ to $\nabla L_{\mathrm{DPO}}(\theta;\mathcal D)+Vx=0$, so the attack perturbs the estimator by adding the fixed gradient shift $Vx$. If $Vx\approx-g^\dagger$, the poisoned optimizer is driven toward the target $\theta^\dagger$.
>
> ## Weakness 1 and Question 3
>
> We respectfully note that only BMP-A is directly inspired by Wen & Li (2021); BAL-A is not derived from that paper. The main novelty of our work is threefold:
>
> - First, we identify the DPO-specific structural property in Section 3 that converts targeted label flipping into a fixed binary sparse approximation problem in gradient space.
> - Second, we design BAL-A through a new binary-aware lattice embedding tailored to the minimum-flip objective and analyze the role of the penalty parameter, including binary-enforcing and minimal-attack recovery.
> - Third, we adapt binary matching pursuit to the *non-normalized* DPO gradient dictionary, and provide DPO-specific attack success guarantees as well as impossibility conditions for this $K$-budgeted attack problem.
>
> So the novelty is not merely modifying an existing algorithm, but revealing the vulnerability of DPO to preference attack due to its geometry, and then designing and analyzing algorithms for that geometry.
>
> ## Limitation
>
> We appreciate the reviewer’s suggestion to add clearer discussion on the limitations of the paper. In the revised paper, we will add the following:
>
> The goal of this paper is to understand and reveal the vulnerabilities of offline RLHF to adversarial preference poisoning, with the hope of raising awareness of this issue in the research community. It is also our intention to design robust RLHF pipelines that can defend against such attacks in future work. As a first step, we restrict the scope of the current paper to a first-principles theoretical study of label flips in the log-linear DPO model, with additional information, e.g., the feature mapping $\psi$, available to the attacker. These assumptions, while providing a theoretically tractable abstraction of general RLHF pipelines, are the main limitations of the paper. Building on this work, we hope to extend the study of attack and defense in more general settings in future work.

---

> > ### Author Rebuttal · Reviewer_DQfh · 2026-04-02
> >
> > Thanks for the answers. The rebuttal is reasonable and helpful. As the authors mentioned, I believe the authors will revise the paper acoordingly. And I keep my score.

---

> > > ### Author Response · Authors · 2026-04-02
> > >
> > > We thank the reviewer again for your efforts in reviewing our paper and for your constructive comments. We are glad to hear that we have adequately addressed all your concerns. We will definitely revise the paper accordingly. We would be sincerely grateful if you could consider updating the score as we have fully addressed your concerns.

---

### Official Review · Reviewer_mfhS · 2026-03-11

**Soundness:** 3
**Presentation:** 2
**Significance:** 3
**Originality:** 3
**Overall Recommendation:** 4
**Confidence:** 1

**Summary:**

This paper studies label-flipping attacks on preference learning methods such as Direct Preference Optimization (DPO). The authors show that the effect of flipping preference labels can be interpreted as manipulating gradient contributions from individual data samples, leading to a formulation of the attack as a sparse vector selection problem that cancels the gradient at a target parameter. Based on this formulation, the paper proposes two algorithms for identifying adversarial flip sets: a lattice-based approach (BAL-A) and a greedy matching pursuit method (BMP-A). Theoretical analysis is provided for both methods, and experiments on synthetic and real datasets demonstrate that a small number of label flips can significantly influence the learned policy.

**Compliance With Llm Reviewing Policy:**

Affirmed.

**Final Justification:**

Overall, the paper presents an interesting and technically grounded study of label-flipping attacks in preference learning, with a particularly appealing formulation that casts the attack as a binary sparse approximation problem in gradient space. The combination of lattice-based and greedy approaches further strengthens the contribution by offering complementary algorithmic perspectives. The authors have provided clear and thoughtful responses that satisfactorily address my concerns. Based on the overall novelty of the formulation, the solid technical contributions, and the authors’ thorough responses, I support the current rating assigned to this paper.

**Key Questions For Authors:**

Please see the Weaknesses.

**Limitations:**

Seems there is no discussion on limitations.

**Strengths And Weaknesses:**

Strengths:
1. The paper provides an interesting perspective on label-flipping attacks in DPO by interpreting them through the lens of gradient manipulation and formulating the problem as a binary sparse approximation problem. This perspective offers a novel way to analyze preference data poisoning.
2. The authors approach the resulting optimization problem from two complementary directions—lattice-based methods and compressed sensing-inspired greedy methods—which provides a convincing and technically grounded framework for constructing attack algorithms.
3. The paper offers helpful explanations regarding the behavior and parameter choices of the proposed algorithms, which improves the interpretability of the methods and aids readers in understanding how the attacks operate in practice.

Weaknesses:
1. The paper evaluates the true positive rate (TPR) by constructing ground-truth flip labels based on Assumption 3.3. However, the exact procedure for constructing this ground-truth set is not clearly described, which makes it difficult to fully verify and interpret the experimental results. A more detailed explanation of how the ground-truth flips are generated would improve the clarity of the evaluation.
2. While the paper demonstrates that preference data poisoning can influence the learned policy, it is difficult to gauge the practical impact of such attacks in real-world scenarios. For example, it would be helpful to include qualitative examples (e.g., changes in LLM responses to representative prompts) to illustrate how the attack affects the model’s behavior.
3. The experimental settings for BAL-A and BMP-A differ substantially, which makes it difficult to directly compare the two approaches. While it is understandable that BAL-A has higher computational complexity, it would strengthen the paper if the authors could provide comparisons under more comparable settings where both methods are feasible.

---

> ### Author Rebuttal · Authors · 2026-03-30
>
> # Reply to Reviewer mfhS
>
> Thank you for the careful review and constructive comments. Below we respond point-by-point and will incorporate these clarifications in the revision.
>
> ## Weakness 1
>
> In each trial, we first sample a support $S\subseteq\{1,\dots,n\}$ with $|S|=K^\star$, define the ground-truth flip vector $x^\star=\mathbb{1}_S$, and then set the target as $t:=Vx^\star=-g^\dagger$, so feasibility is enforced by construction. The reported TPR is then computed against the sampled support $S$.
>
> ## Weakness 2
>
> We agree with the reviewer that qualitative examples like changed LLM responses would be useful for illustrating real-world impact. However, such evaluations depend heavily on the choice of model, target behavior, and prompting setup, and would require substantially more engineering and computation than the current theoretical study. Moreover, extending the attack formulation to a more general real-world RLHF pipeline for LLMs is possible but introduces extra challenges (as discussed in the reply to Question 1 of reviewer **m3ed**). Therefore, we restrict the current experiments to validating attack on log-linear DPO through controlled simulations.
>
> To better illustrate how the attack affects the model’s behavior, we additionally include an explicit clean-vs-attacked comparison between training on the *clean* SHP subset $\mathcal D$ and training on the *attacked* SHP subset $\tilde{\mathcal D}(\hat x)$ produced by BAL-A/BMP-A, using the same downstream diagnostics as in the paper (parameter/policy distance).
>
> We use the SHP subset with $n = 50$ and $K^\star=7$ (the same as Appendix I), and run BAL-A with $M=0.9$ and BMP-A with $K_{\max}=7$. In this setup, both methods exactly recover the same ground-truth flip pattern ($\mathrm{TP}=7,\ \mathrm{TPR}=1.0$), so the attacked dataset is identical for the two methods.
>
> Retraining log-linear DPO on the attacked subset (learned $\pi_{\hat{\theta}}$) yields a *clear downstream deviation* from training on the clean subset (learned $\pi_{\text{clean}}$), with $| \pi_{\text{clean}} - \pi_{\hat{\theta}} |$ increasing from 0 at initialization to about 0.224 by step 20000. Meanwhile, for the target policy $\pi_{\theta^\dagger}$, $| \pi_{\theta^\dagger} - \pi_{\hat{\theta}} |$ is reported around 0.012 in Appendix I, which is only about 5\% of the clean-policy gap. Thus, BAL-A/BMP-A mislead the learned downstream policy much closer to the target than to the clean policy. We include the table of policy distances here for reference, and will add the figures for parameter/policy distances and discussion in the revised paper.
>
> | Optimization step | Clean vs. BAL-A | Clean vs. BMP-A |
> | - | - | - |
> | 0 | 0.0000 | 0.0000 |
> | 100 | 0.0760 | 0.0760 |
> | 500 | 0.1376 | 0.1376 |
> | 1000 | 0.1580 | 0.1580 |
> | 5000 | 0.2019 | 0.2019 |
> | 10000 | 0.2148 | 0.2148 |
> | 20000 | 0.2236 | 0.2236 |
>
> ## Weakness 3
>
> We note that the two methods are designed for different attack formulations: BAL-A targets the minimum-flip attack, while BMP-A targets the $K$-sparse attack. For this reason, our current experiments emphasize the problem in which each method is most appropriate, rather than presenting them as two interchangeable solvers for the same problem.
>
> However, we fully agree with the reviewer that a side-by-side comparison under a common feasible setting would be informative. Therefore, we additionally include experiments on a smaller and common setup where both methods can be run with the same underlying data and comparable parameters, and we compare the attack-pattern recovery of both methods using the same recovery metric as in the paper (support TPR). We use this additional experiment to clarify their empirical behavior under matched conditions, while keeping the main text focused on the distinct attack settings each method is intended to solve.
>
> We run BAL-A and BMP-A on one common SHP subset with $n = 50$ and $K^\star=7$ (the same setup as in Weakness 2). In this matched setup, both methods exactly recover the same ground-truth flip pattern with $\mathrm{TP}=7,\ \mathrm{FP}=0,\ \mathrm{FN}=0$ and $\mathrm{TPR}=1.0$. Therefore, both BAL-A and BMP-A successfully recover the attack pattern in this comparable setting.
>
> As for the higher computational complexity of BAL-A, the dominant cost is the lattice preprocessing on the $(d+n)\times n$ embedded basis, namely LLL reduction together with the linear-algebra preprocessing. Babai itself is comparatively cheap once the reduced basis is available. Thus, BAL-A is best viewed as a method for small-to-moderate attacked subsets, while the larger-scale budgeted attack problem is handled by BMP-A. For this comparable small simulation, the runtime of BAL-A is about 0.68653 seconds, while the runtime of BMP-A is about 0.00013 seconds.
>
> ## Limitation
>
> In revision, we will add the discussion of limitations (details in Reply to Limitation of reviewer **DQfh**).

---

> > ### Author Rebuttal · Reviewer_mfhS · 2026-04-01
> >
> > The authors have provided clear and thoughtful responses to the reviewer’s concerns. The explanations are helpful and make the experimental setup and results easier to understand. Overall, the responses are well-articulated, and the reviewer finds them reasonable and satisfactory.

---

> > > ### Author Response · Authors · 2026-04-02
> > >
> > > We thank the reviewer again for your efforts in reviewing our paper and for your constructive comments. We are glad to hear that we have adequately addressed all your concerns. We would be sincerely grateful if you could consider updating the score to reflect this.

---

### Official Review · Reviewer_m3ed · 2026-03-13

**Soundness:** 2
**Presentation:** 3
**Significance:** 2
**Originality:** 4
**Overall Recommendation:** 4
**Confidence:** 3

**Summary:**

This paper provides the first theoretical analysis of label flip attacks against the Direct Preference Optimization (DPO) pipeline in offline Reinforcement Learning from Human Feedback (RLHF). The authors identify a "simple but powerful" structural property: in log-linear DPO, flipping a preference label induces a gradient shift that is independent of the current policy parameters. This observation allows them to reformulate a targeted poisoning attack as a binary sparse approximation problem. To solve this, the paper introduces two novel algorithms:Binary-Aware Lattice Attack (BAL-A): Uses lattice embedding and LLL reduction to recover minimum-flip patterns.Binary Matching Pursuit Attack (BMP-A): A greedy approach for budgeted flip scenarios with coherence-based recovery guarantees.
The theory is validated through synthetic experiments and the Stanford Human Preferences (SHP) dataset.

**Compliance With Llm Reviewing Policy:**

Affirmed.

**Final Justification:**

The rebuttal adequately addressed my concerns. I am satisfied with the clarifications and will update my recommendation to accept.

**Key Questions For Authors:**

1. Do the authors expect this structural property to hold approximately for neural network policies used in practical RLHF settings?
2. So, following this point, can the authors provide empirical evidence showing that the proposed attacks remain effective when applied to neural DPO training?

3. Can the authors demonstrate the effect of the recovered flip patterns on an actual RLHF training pipeline (e.g., training a model with DPO on a dataset containing the attacked labels and evaluating downstream policy behavior)?

4. Could the authors clarify the computational complexity of BAL-A and discuss whether it scales to preference datasets of realistic size (e.g., hundreds of thousands of comparisons)?

**Limitations:**

The paper could benefit from a clearer discussion of (i) the assumptions required for the attacks (e.g., log-linear policies and white-box access), (ii) the gap between the theoretical model and realistic RLHF pipelines, and (iii) potential defensive implications or mitigation strategies.

**Strengths And Weaknesses:**

Strengths

The paper identifies an interesting structural property of log-linear DPO: flipping a single preference label produces a parameter-independent gradient shift. This insight enables the poisoning attack problem to be reformulated in gradient space. It simplifies the analysis and leads to a clean mathematical formulation. Furthermore, the reduction of the targeted label-flip attack problem to a binary sparse approximation problem is conceptually interesting. This perspective connects RLHF security analysis with classical sparse recovery and signal processing tools. To address this, the paper proposes two attack algorithms addressing different regimes: BAL-A for the minimum-flip objective using lattice methods, and BMP-A for the budgeted sparse attack scenario.


Weakness
The analysis relies heavily on the log-linear policy class, which significantly simplifies the structure of the DPO gradient. Modern RLHF pipelines typically involve large neural networks rather than log-linear policies, and it remains unclear whether the key structural property of parameter-independent gradient shifts holds approximately in these settings. The evaluation focuses on recovering flip supports rather than demonstrating practical degradation in policy behavior, and no experiments are conducted on actual RLHF training pipelines or LLM fine-tuning. Consequently, the practical impact of these attacks on realistic RLHF systems remains unclear, especially since the work focuses only on label-flip attacks while other poisoning strategies, such as data injection or reward manipulation, may be more realistic in practical RLHF pipelines. Furthermore, BAL-A relies on lattice reduction techniques. The computational complexity can become significant for large datasets, and the attack requires white-box knowledge of the feature mapping $\psi$ and the reference policy $\pi_{ref}$ to compute the target gradient $g^{\dagger}$, and it may cause to limit the immediate threat in proprietary LLM settings. The analysis relies heavily on the log-linear policy class, which significantly simplifies the structure of the DPO gradient. While this assumption is common in theoretical RLHF work to isolate core geometric phenomena, modern RLHF pipelines typically involve large neural networks rather than log-linear policies. It is therefore unclear whether the key structural property—parameter-independent gradient shifts—holds even approximately in these high-dimensional, non-linear settings.Furthermore, the experiments primarily validate the recovery behavior of the proposed algorithms on synthetic dictionaries and gradient dictionaries constructed from the Stanford Human Preferences (SHP) dataset. However, several factors limit the soundness of the practical claims:Support Recovery vs. Policy Impact: The evaluation focuses largely on recovering flip supports (True Positive Rate) rather than demonstrating a tangible, practical degradation in policy behavior or safety.Lack of End-to-End Training: No experiments are conducted on actual large-scale RLHF training pipelines or LLM fine-tuning; the results are mostly confined to gradient-space approximations.Scope of Attack: The work focuses exclusively on label-flip attacks. In practical RLHF pipelines, other poisoning strategies—such as data injection or reward manipulation—might be more realistic or impactful.White-Box Assumptions: The current attack requires explicit knowledge of the feature mapping $\psi$ and the reference policy $\pi_{ref}$ to compute the target gradient $g^{\dagger}$. In proprietary or "black-box" LLM settings, this requirement significantly limits the immediate threat of these specific algorithms.Computational Scalability: BAL-A relies on lattice reduction techniques like LLL, the complexity of which becomes significant as the dataset size $n$ grows. It remains unproven whether this approach can scale to preference datasets containing hundreds of thousands or millions of comparisons.Consequently, while the paper is theoretically rigorous within its defined constraints, the practical impact and generalizability of these attacks on realistic, non-linear RLHF systems remain an open question.

---

> ### Author Rebuttal · Authors · 2026-03-30
>
> # Reply to Reviewer m3ed
>
> Thank you for constructive comments. Below we respond point-by-point and will incorporate these clarifications in the revision.
>
> ## Question 1
>
> We do not expect the parameter-independent gradient-shift identity to hold exactly for general neural network policies. In DPO with general policy, flipping one label induces
> $$\nabla_\theta \ell_\theta^{\text{flip}}(s,a_w,a_l)-\nabla_\theta \ell_\theta(s,a_w,a_l)=\beta\big(\nabla_\theta\log\pi_\theta(a_w\mid s)-\nabla_\theta\log\pi_\theta(a_l\mid s)\big),$$
> which is typically $\theta$-dependent. Thus, the $\theta$-independent shift in Theorem 3.1 is a structural property of log-linear policy.
>
> However, part of the attack formulation can still be extended to general policy *locally* at the target policy parameter $\theta^\dagger$. For each pair $(s^i, a_w^i, a_l^i)$, define
> $$v_i (\theta^\dagger):=\beta\big(\nabla_\theta\log\pi_\theta (a_w^i \mid s^i)-\nabla_\theta\log\pi_\theta(a_l^i\mid s^i)\big)\big|\_{\theta^\dagger}, \quad V\_{\theta^\dagger}:=[v_1(\theta^\dagger),\dots,v_n(\theta^\dagger)].$$
> Consider
> $$\min_{x\in\{0,1\}^n}\ \mathbf{1}^\top x\qquad\text{s.t.}\qquad V\_{\theta^\dagger}x=-g^\dagger.$$
> So BAL-A/BMP-A may still be used as solvers for this local binary sparse-approximation problem.
>
> Compared with log-linear case, the main challenges are:
>
> - computing $V_{\theta^\dagger}$ requires per-sample backpropagation and is more expensive in large neural models.
> - $V_{\theta^\dagger}x=-g^\dagger$ is only a *local first-order condition* around $\theta^\dagger$, not the global optimum in Section 3.2.
> - neural DPO objective is generally nonconvex, so gradient-space attack effect no longer directly implies parameter/policy-space guarantees of Lemma 3.2.
>
> ## Question 2
>
> As in Question 1, it is possible to extend the proposed attacks to neural DPO, with extra challenges and additional computational and engineering efforts. Therefore, we humbly restrict the current scope to label flips on log-linear DPO, though the attack idea can be extended. We view this as important future work.
>
> ## Question 3
>
> The training pipeline in the paper's scope is log-linear DPO. In Appendix I, we report attacked-vs-groundtruth downstream policy comparison on $\tilde{\mathcal D}(\hat x)$ and $\tilde{\mathcal D}(x^\star)$, which verifies that attacks produce the targeted attack effect. We agree and acknowlegde, however, that a clean-vs-attacked comparison on $\mathcal D$ and $\tilde{\mathcal D}(\hat x)$ by BAL-A/BMP-A makes the effect of attack clearer.
>
> We run additional experiment to include this comparison using the same downstream metrics (parameter/policy distance). Detailed setup and policy-distance result table is given in Reply to Weakness 2 of reviewer **mfhS**.
>
> Retraining on the attacked subset yields a *clear downstream deviation* from retraining on the clean subset: $| \pi_{\text{clean}} - \pi_{\hat{\theta}} | \approx 0.224$. By contrast, Appendix I reports $| \pi_{\theta^\dagger} - \pi_{\hat{\theta}} | \approx 0.012$, which is only about 5\% of the clean-policy gap. Thus the attacks move the learned policy much closer to the target than to the clean policy.
>
> ## Question 4
>
> The dominant cost of BAL-A is the lattice preprocessing on the $(d+n)\times n$ embedded basis (LLL reduction plus linear-algebra preprocessing). Babai itself is relatively cheap once the reduced basis is available. Thus BAL-A is best suited to small-to-moderate attacked subsets. That is why our experiments use BAL-A on smaller SHP subsets and BMP-A for larger-scale budgeted attacks.
>
> For the added experiment above, the runtime of BAL-A is around 0.68653 sec, while BMP-A takes around 0.00013 sec. Both methods scale to larger datasets, but BMP-A handles this better. We also note that the exact separation-threshold computation in Theorem 4.3 is combinatorial and serves only as a sufficient condition for theretical validation, not as part of the scalable attack pipeline.
>
> ## Weakness of Scope of Attack
>
> Data injection is complementary and has been studied in prior work. We note that Nika et al. (2025) shows that misleading RLHF by data injection requires appending numerous poisoned preference pairs, which makes label flip attack more practically plausible.
>
> Reward manipulation is an open direction. For a two-stage RLHF in Appendix B, manipulating the reward is not enough; one must still analyze how the poisoned reward propagates through policy optimization stage. This extra stage is one reason why we focus on DPO: it gives a direct objective over the policy parameter, allowing cleaner theory. We also note that in RLHF with linear reward, label flip attack also induces *parameter-independent gradient shift* $\Delta\psi$ (details in Reply to Question 5 of reviewer **cBSx**). The remaining challenge is the two-stage propagation, which is important future work.
>
> ## Limitation
>
> In revision, we will add the discussion of limitations (details in Reply to Limitation of reviewer **DQfh**).

---

> > ### Author Rebuttal · Reviewer_m3ed · 2026-04-03
> >
> > The rebuttal adequately addressed my concerns. I am satisfied with the clarifications and will update my recommendation to accept

---

> > > ### Author Response · Authors · 2026-04-03
> > >
> > > We thank the reviewer again for your efforts in reviewing our paper and for your constructive comments. We are glad to hear that we have adequately addressed all your concerns and that the reviewer has raised the score. We are very grateful for the reviewer's support.

---

### Decision · Program_Chairs · 2026-04-30

**Decision:**

Accept (regular)

**Comment:**

This paper studies label flip attacks under preference optimization and builds a connection to sparse approximation. The insight is interesting, and reviewers are convinced of the effectiveness of the algorithms. After author rebuttal, no major concern remains.